# Fast Adversarial Robustness Certification of Nearest Prototype Classifiers for Arbitrary Seminorms

**Sascha Saralajew**[*]
Dr. Ing. h.c. F. Porsche AG
Weissach, Germany
`sascha.saralajew@porsche.de`

**Lars Holdijk**[*]
University of Amsterdam
Amsterdam, Netherlands
`larsholdijk@gmail.com`

**Thomas Villmann**
UAS Mittweida
Mittweida, Germany,
`villmann@hs-mittweida.de`

## Abstract

Methods for adversarial robustness certification aim to provide an upper bound on the test error of a classifier under adversarial manipulation of its input. Current certification methods are computationally expensive and limited to attacks that optimize the manipulation with respect to a norm. We overcome these limitations by investigating the robustness properties of Nearest Prototype Classifiers (NPCs) like learning vector quantization and large margin nearest neighbor. For this purpose, we study the hypothesis margin. We prove that if NPCs use a dissimilarity measure induced by a seminorm, the hypothesis margin is a tight lower bound on the size of adversarial attacks and can be calculated in constant time—this provides the first adversarial robustness certificate calculable in reasonable time. Finally, we show that each NPC trained by a triplet loss maximizes the hypothesis margin and is therefore optimized for adversarial robustness. In the presented evaluation, we demonstrate that NPCs optimized for adversarial robustness are competitive with state-of-the-art methods and set a new benchmark with respect to computational complexity for robustness certification.

## 1 Introduction

Adversarial robustness of a classifier describes its stability in classification under adversarial manipulations of the input. The adversarial setting has been studied extensively in numerous settings [1–3] but mainly found footing after the seminal paper by Szegedy et al. [4] that formalized the problem of *adversarial examples*. Since that, a wide line of research has sprung concerning both the construction of adversarial attacks [5–7] and the heuristic defense against them [8–10]. Unfortunately, while some progress has been made [11–13], most proposed defenses have been shown to be breakable by more advanced attacks [14–16] so that the adversarial robustness problem is far from being solved. With the heuristic defenses at the losing side of the metaphorical arms race, provable robustness guarantees for classifiers provide a welcome alternative [11, 12, 17]. Robustness guarantees aim to provide the so-called *robust test error* (or an upper bound) of a classifier under adversarial attacks and are therefore not dependent on the current state of adversarial attacks. However, current methods for determining the robust test error are computationally expensive [12, 17, 18] and are often limited to $L^p$-norm evaluations [11, 13, 18]. As an adverse effect, we cannot use them for the rejection of adversarial examples regarding an arbitrary seminorm without a huge computational overhead.

---

[*]Authors contributed equally.

To tackle these limitations, we extend the study of robustness guarantees beyond Neural Networks (NNs), on which most work has focused, by investigating *Nearest Prototype Classifiers* (NPCs) [19–22]. For that, given a data space $\mathcal{X}$, an NPC is defined by a set $\mathcal{W}$ of *prototypes* selected from the data space $\mathcal{X}$ and a dissimilarity measure $d : \mathcal{X} \times \mathcal{X} \to \mathbb{R}_{\geq 0}$. Each prototype $\mathbf{w} \in \mathcal{W}$ has a predefined and fixed class label $c(\mathbf{w}) \in \mathcal{C} = \{1, 2, \ldots, N_c\}$. The class assignment $c^*_{\mathcal{W}}(\mathbf{x}) \in \mathcal{C}$ of a data point $\mathbf{x} \in \mathcal{X}$ is determined by the class label of the closest prototype $\mathbf{w}^*$ (1-nearest neighbor rule):

$$c^*_{\mathcal{W}}(\mathbf{x}) = c(\mathbf{w}^*) \text{ with } \mathbf{w}^* = \argmin_{\mathbf{w} \in \mathcal{W}} \{d(\mathbf{x}, \mathbf{w})\}. \tag{1}$$

Due to the fundamental interpretability of the prototypes and dissimilarity measure used, NPCs are considered to be among the most interpretable machine learning models. This makes NPCs a preferred choice in the medical field, where the interpretability of models is a requirement for clinical trials [23, 24]. As a result, prototype-based principles have been adopted in a number of deep learning fields. Amongst the most notable of those is few-shot learning [25]. In addition to this, an empirical study has shown that NPCs are robust against adversarial attacks [26]. In summary—with the call for interpretable machine learning models increasing, the NPC principles adopted in deep learning, and the promising empirical robustness results—NPCs provide a worthwhile avenue for studying guaranteed adversarial robustness.

**Contributions** We analyze the adversarial robustness properties of NPCs in terms of the hypothesis margin. First, we show that if the dissimilarity measure $d(\mathbf{x}, \mathbf{w})$ is induced by a seminorm $\|\cdot\|$ (i. e., $d(\mathbf{x}, \mathbf{w}) = \|\mathbf{x} - \mathbf{w}\|$), the hypothesis margin regarding this seminorm can be computed in constant time during the inference—this even holds in the case of an uncountable set of prototypes. Second, we prove that this margin is a tight lower bound on the magnitude of an adversarial attack measured by the same seminorm—to the best of our knowledge, this presents the first robustness guarantee that holds for an arbitrary seminorm. Third, using this result, we show that every NPC that classifies by a seminorm and is trained by minimizing a triplet loss is inherently optimizing the adversarial robustness with respect to this seminorm. In an experimental section, we present how these results apply to different NPCs and that a violation of the assumptions (seminorm and triplet loss) does not necessarily lead to adversarially robust methods. The experimental results highlight that the derived robustness certificate is comparable with other methods in terms of guaranteed robustness but outperforms them all when computation speed is considered.

The following section discusses related work. After that, we give a brief introduction to NPCs and define the models we use in the evaluation. This section is followed by the main part where we define the hypothesis margin and investigate the relation to adversarial robustness. The subsequent experimental section shows how to apply these results and compares the derived robustness certificate to other methods. Finally, we finish with a discussion and an outlook of the presented results.

## 2    Related work

Besides general theoretical work about adversarial robustness [27–30], the defense investigations can be grouped into three areas: *robustification*, the research to improve the adversarial robustness of models [8, 10, 11, 31–33]; *verification* (complete or exact methods), the analysis of how to compute the robustness guarantees exact [17, 34, 35]; *certification* (incomplete methods), the study of fast calculable bounds for the robustness guarantees [36–38]. It is important to distinguish between verification methods and certification approaches. Naturally, verification approaches [17] have a combinatorial time complexity even though there are attempts to improve the computational complexity [39]. In contrast, certification methods [12, 40] try to return bounds for the robustness guarantees in polynomial time. Comparing verification and certification results only in terms of the returned robustness guarantees ignores the aspect of time complexity of the methods and is therefore not a fair comparison. We consider this thought carefully throughout the evaluation.

The presented work focuses on both the study of fast calculable certificates for NPCs and the use of the certificate to robustify the models. Currently, the most successful approach to robustify models is adversarial training [4, 41, 42]. However, being dependent on an adversarial attack used during training, adversarial training fails to provide a robustness certificate and increases the training time. Other empirical robustification approaches [8, 9, 43, 44] struggle with similar problems or fail to withstand stronger attacks [5, 7, 14–16, 41, 45–47]. It is therefore not surprising that several modern approaches strive—as the work presented in this article—towards combining robustification and

certification. Wong et al. [12, 40] studied an approach for NNs based on *convex outer adversarial polytopes*. Their method extends to arbitrary norms but is still computationally expensive and infeasibly complex to compute for deep NNs. Randomized Smoothing yields another approach mainly investigated for NNs despite being applicable for *arbitrary* classifiers [18, 48, 49]. This approach fails to scale to arbitrary norms though and introduces a heavy computational overhead through sampling. Besides, some work has also been done to study certification and verification for other classification approaches like decision trees [50, 51] and support vector machines [2, 36, 52] or to extend robustness certificates to an arbitrary $L^p$-norm knowing the results for a few $L^p$-norms [13].

Based on empirical observations, Saralajew et al. [26] discussed the relation between the margin maximization properties of Generalized Learning Vector Quantization (GLVQ) [53] and its adversarial robustness *without* providing a mathematical proof. The presented results rely on the hypothesis margin maximization properties of Learning Vector Quantization (LVQ) [54, 55], as originally studied by Crammer et al. [56]. There, the hypothesis margin was used to derive a generalization bound for LVQ with the Euclidean norm. Consequently, these results are not applicable to seminorms or an uncountable set of prototypes in arbitrary NPCs. Similarly, the work of Wang et al. [57] about the adversarial robustness of k-nearest neighbors cannot be scaled to an arbitrary NPC even though the results hold for an arbitrary norm. Wang et al. [58] presented a result that is similar to the hypothesis margin but limited to $L^1$-, $L^2$-, and $L^\infty$-norms for the attack and $L^2$-norms for the classifier metric. However, their methods are orders of magnitude slower. Besides that, Yang et al. [59] proposed a generic defense by preprocessing the dataset before training a k-nearest neighbors classifier with adversarial pruning. In general, the method can be used to robustify NPCs, but for more than two classes it is time-consuming and it cannot be used to robustify seminorm-based NPCs. Brinkrolf et al. [60] studied Generalized Matrix LVQ (GMLVQ) [61] with reject options and derived an adversarial perturbation bound needed to fool the classifier. Compared to our work, the method requires the training of the classifier with reject options and does not provide a general framework for the evaluation of adversarial robustness and the robustification of NPCs.

## 3 Nearest prototype classifiers

As already mentioned in the introduction, an NPC consists of two main building blocks: a set $\mathcal{W}$ of prototypes and a dissimilarity measure $d$. The prototypes $\mathbf{w} \in \mathcal{W}$ are elements of the data space $\mathcal{X}$ and have a predefined class label $c(\mathbf{w}) \in \mathcal{C}$. Given a prototype $\mathbf{w}$ and an input sample $\mathbf{x}$, we compute the "distance" between these two elements by the dissimilarity $d$. Based on this distance, we assign the class of the closest prototype $\mathbf{w}^*$ to a given input, see Equation (1). The closest decision boundary to a given input $\mathbf{x}$ is *implicitly* defined by the closest prototype $\mathbf{w}^*$ and the closest prototype $\mathbf{w}_*$ with a *different* class label than $\mathbf{w}^*$ based on the given dissimilarity $d$.

In an NPC, the determination of the closest prototype of each class is considered as the *inference step*. During training, an NPC updates the prototypes and the maybe trainable dissimilarity measure. The training can be realized by heuristic methods and loss-based optimization and is an important difference between different realizations. For example, some NPCs optimize the selection of prototypes out of a given set of labeled data points (e. g., 1-nearest neighbor) or optimize the prototypes as free parameters (e. g., LVQ). Following the empirical observation of Saralajew et al. [26], we focus our analysis on the family of LVQ algorithms. These methods not only train the dissimilarity measure but the prototypes as well. In particular, compared to 1-nearest neighbor methods, LVQ does not only optimize the selection of prototypes out of a given training dataset but instead considers the prototypes as fully trainable parameters. We refer to the articles of Biehl et al. [22] and Nova and Estévez [21] for an overview and an in-depth introduction to LVQ and NPCs.

**Generalized learning vector quantization** GLVQ provides an LVQ version that can be trained by an empirical risk minimization and satisfies the convergence condition [53]. Because the method is trained by a gradient-based approach, the dissimilarity measure has to be differentiable almost everywhere. Except for this condition, the dissimilarity measure can be chosen freely (e. g., squared Euclidean distance). Given a training dataset $\mathcal{T}$ of labeled inputs $(\mathbf{x}, c(\mathbf{x}))$, we fix the number of prototypes per class, initialize the prototypes, and optimize the prototypes by minimizing the following averaged loss function:

$$\frac{1}{\#\mathcal{T}} \sum_{(\mathbf{x}, c(\mathbf{x})) \in \mathcal{T}} \mu(\mathbf{x}, c(\mathbf{x})) = \frac{1}{\#\mathcal{T}} \sum_{(\mathbf{x}, c(\mathbf{x})) \in \mathcal{T}} \frac{d(\mathbf{x}, \mathbf{w}^+) - d(\mathbf{x}, \mathbf{w}^-)}{d(\mathbf{x}, \mathbf{w}^+) + d(\mathbf{x}, \mathbf{w}^-)}, \tag{2}$$

where $\mathbf{w}^+$ is the closest prototype to $\mathbf{x}$ of the correct class $c(\mathbf{x})$ with respect to $d$ and $\mathbf{w}^-$ is the closest prototype to $\mathbf{x}$ of an incorrect class. The expression $\mu(\mathbf{x}, c(\mathbf{x})) \in [-1, 1]$ is called the *relative distance difference* and returns negative values if and only if $\mathbf{x}$ is correctly classified.

**Generalized tangent learning vector quantization**   The Generalized Tangent LVQ (GTLVQ) algorithm is a version of GLVQ where a tangent distance is used [62]. The prototypes in GTLVQ are defined as elements of affine subspaces of the data space $\mathcal{X} = \mathbb{R}^n$. In other words, we can consider GTLVQ as an NPCs with infinitely many prototypes, approximating variations within the classes. The dissimilarity of a given data point $\mathbf{x}$ to the $k$-th affine subspace is measured in terms of the smallest Euclidean distance $d_E$:

$$\min \{ d_E(\mathbf{x}, \mathbf{t}_k + \mathbf{B}_k \boldsymbol{\theta}) \mid \boldsymbol{\theta} \in \mathbb{R}^m \} = d_E\left(\mathbf{x}, \mathbf{t}_k + \mathbf{B}_k \mathbf{B}_k^{\mathrm{T}}(\mathbf{x} - \mathbf{t}_k)\right), \tag{3}$$

where $\mathbf{B}_k \in \mathbb{R}^{n \times m}$ is an $m$-dimensional orthonormal basis, $\mathbf{t}_k \in \mathbb{R}^n$ is a translation vector, and $\mathbf{t}_k + \mathbf{B}_k \mathbf{B}_k^{\mathrm{T}}(\mathbf{x} - \mathbf{t}_k)$ determines the closest prototype at the $k$-th affine subspace. Given the number of affine subspaces and the dimension $m$, we optimize the bases $\mathbf{B}_k$ and translations $\mathbf{t}_k$ by minimizing Equation (2) with Equation (3) as the dissimilarity measure during training.

**Robust soft learning vector quantization**   Robust Soft LVQ (RSLVQ) is a probabilistic version of LVQ where the prototypes are assumed as centers in a Gaussian mixture model [63].[2] The posterior probability $P(l \mid \mathbf{x})$ that an input $\mathbf{x}$ belongs to a certain class $l \in \mathcal{C}$ is computed by

$$P(l \mid \mathbf{x}) = \frac{\sum_{\mathbf{w}:c(\mathbf{w})=l} \exp\left(-d_E^2(\mathbf{x}, \mathbf{w})\right)}{\sum_{\mathbf{w}} \exp\left(-d_E^2(\mathbf{x}, \mathbf{w})\right)}. \tag{4}$$

The prototypes are determined by minimizing the cross-entropy loss between the predicted probability vector $\mathbf{p}(\mathbf{x}) = (P(1 \mid \mathbf{x}), \ldots, P(N_c \mid \mathbf{x}))^{\mathrm{T}}$ and the true probability vector (one-hot encoding) of a labeled input $(\mathbf{x}, c(\mathbf{x}))$. Considering Equation (4), we observe that the calculation of the probabilities follows a softmax squashing. Together with the cross-entropy loss, this makes the RSLVQ algorithm highly similar to the design and training of classification layers in NNs.

# 4   Hypothesis margin maximization and adversarial robustness certification

In this section, we define the hypothesis margin for NPCs and derive some of its properties. For instance, we show that the hypothesis margin can be easily computed, lower bounds adversarial perturbations, and can be optimized during training. We refer the reader to Section A of the supplementary material for a visualization of the defined concepts and their properties in $\mathbb{R}^2$.

## 4.1   Definition and calculation of the hypothesis margin

Margins are a common tool to measure the confidence of a classifiers decision. For example, the sample margin—the distance of a sample to the closest decision boundary—is used in the optimization of support vector machines. In the context of NPCs, the sample margin is defined as follows.

**Definition 1** (sample margin)**.** Given a set $\mathcal{W}$ of prototypes and a dissimilarity $d$. The *sample margin* of $\mathcal{W}$ with respect to a set $\mathcal{S}$ of inputs is the maximum radius $r$ such that the following condition holds: If we define a ball with radius $r$ induced by $d$ around each sample $\mathbf{x}$ of $\mathcal{S}$, each point within a ball has the same assigned class label as the center $\mathbf{x}$ of the ball. In symbols, we write $\mathrm{margin}_s(\mathcal{S}, \mathcal{W})$.

As a decision boundary is implicitly defined by two prototypes of different classes, this margin definition is cumbersome for NPCs. In particular, given two prototypes $\mathbf{w}_i$ and $\mathbf{w}_j$ of different classes, a decision boundary is defined by those elements $\mathbf{x} \in \mathcal{X}$ for which $d(\mathbf{x}, \mathbf{w}_i) = d(\mathbf{x}, \mathbf{w}_j)$. Therefore, depending on the dissimilarity $d$, the calculation of the decision boundary can be difficult. The *hypothesis margin* offers an alternative and more suitable concept for NPCs.

**Definition 2** (hypothesis margin). Given a set $\mathcal{W}$ of prototypes and a dissimilarity $d$. The *hypothesis margin* of $\mathcal{W}$ with respect to a set $\mathcal{S}$ of inputs is the maximum radius $r$ such that the following condition holds: If we define a ball with radius $r$ induced by $d$ around each prototype, every change in the position of the prototypes within its ball does not change the class labels assigned to the inputs of $\mathcal{S}$. In symbols, we write $\mathrm{margin}_h\left(\mathcal{S},\mathcal{W}\right)$.

We assume in these definitions—and in general—that the class label assignments to $\mathcal{S}$ by $\mathcal{W}$ are *unambiguous* and well-defined. This means that input samples do not lie on a decision boundary as otherwise the class assignments would be ill-defined. Additionally, a ball is always assumed as an *open* set. With the class assignments considered as unambiguous, this is not a limitation.

With the next theorem, we provide a formula to compute the hypothesis margin of a sample using only one additional floating-point operation after the inference of the NPC. This theorem is valid for seminorms and, thus, holds for all NPCs where the dissimilarity is induced by a seminorm.

**Definition 3** (seminorm). Given a vector space $\mathcal{V}$ over a field $\mathcal{F}$ of the real or complex numbers, a *seminorm* $\|\cdot\|$ is a function $\|\cdot\| : \mathcal{V} \longrightarrow \mathbb{R}$ that satisfies the following conditions for all $\mathbf{x}, \mathbf{y} \in \mathcal{V}$ and $\alpha \in \mathcal{F}$: $\|\mathbf{x}\| \geq 0$ (nonnegativity); $\|\alpha\mathbf{x}\| = |\alpha| \|\mathbf{x}\|$ (absolute homogeneity); $\|\mathbf{x} + \mathbf{y}\| \leq \|\mathbf{x}\| + \|\mathbf{y}\|$ (triangle inequality).

**Theorem 1.** *Let the data space $\mathcal{X}$ be a vector space over a field of the real or complex numbers, $d\left(\mathbf{x},\mathbf{w}\right) = \|\mathbf{x} - \mathbf{w}\|$ be a dissimilarity induced by a seminorm $\|\cdot\|$, and $\mathbf{x} \in \mathcal{X}$ be an input. Then, the hypothesis margin of the set $\mathcal{W}$ of prototypes with respect to $\mathbf{x}$ can be computed by*

$$\mathrm{margin}_h\left(\{\mathbf{x}\},\mathcal{W}\right) = \frac{1}{2}\left(\|\mathbf{x} - \mathbf{w}_*\| - \|\mathbf{x} - \mathbf{w}^*\|\right), \tag{5}$$

*where $\mathbf{w}^*$ denotes the closest prototype to $\mathbf{x}$ and $\mathbf{w}_*$ denotes the closest prototype to $\mathbf{x}$ with a different class label than the class label of $\mathbf{w}^*$.*

A proof of the theorem can be found in the supplementary material Section B. Based on this theorem, we have a surprisingly simple rule to compute the hypothesis margin after the inference with only one additional floating-point operation and, therefore, in constant time.

## 4.2 Hypothesis margin lower bounds the adversarial perturbation

Given an input sample $\mathbf{x} \in \mathcal{X}$ with the class label $c\left(\mathbf{x}\right)$, an *adversarial example* $\tilde{\mathbf{x}}$ of the sample $\mathbf{x}$ is defined by an *adversarial perturbation* $\boldsymbol{\delta}$ of $\mathbf{x}$ such that $\tilde{\mathbf{x}} = \mathbf{x} + \boldsymbol{\delta}$ is a point on the decision boundary or in the classification region of a different class than $c\left(\mathbf{x}\right)$. Frequently, the computation of an adversarial perturbation is treated as an optimization problem by searching for a perturbation with minimum magnitude:

$$\min_{\boldsymbol{\delta}} \|\boldsymbol{\delta}\| \text{ such that } c_{\mathcal{W}}^*\left(\tilde{\mathbf{x}}\right) \neq c\left(\mathbf{x}\right) \text{ and } \tilde{\mathbf{x}} = \mathbf{x} + \boldsymbol{\delta} \in \mathcal{X}. \tag{6}$$

We should note that the magnitude of the adversarial perturbation is measured in terms of a seminorm $\|\cdot\|$ and that the adversarial example has to be an element of the input space $\mathcal{X}$. Sometimes additional conditions are placed upon the adversarial example (e. g., it has to be a sample of a certain class, an element of a subset of $\mathcal{X}$, etc.). Frequently, the optimization task of Equation (6) is intractable and, hence, the optimal perturbation can only be approximated. A procedure that generates adversarial examples is called an *adversarial attack*. An adversarial attack that only creates adversarial examples with a maximum perturbation $\|\boldsymbol{\delta}\|$ less than or equal a given bound $\epsilon > 0$ is called $\epsilon$-*limited adversarial attack*.

The next theorem and corollary provide a statement about the relation between the sample and the hypothesis margin—see Section C of the supplementary material for the proofs.

**Theorem 2.** *Let the data space $\mathcal{X}$ be a vector space over a field of the real or complex numbers, $d\left(\mathbf{x},\mathbf{w}\right) = \|\mathbf{x} - \mathbf{w}\|$ be a dissimilarity induced by a seminorm $\|\cdot\|$, and $\mathcal{S}$ be a set of inputs. Then, the hypothesis margin of $\mathcal{W}$ with respect to $\mathcal{S}$ yields a lower bound on the sample margin of $\mathcal{W}$ with respect to $\mathcal{S}$:*

$$\mathrm{margin}_h\left(\mathcal{S},\mathcal{W}\right) \leq \mathrm{margin}_s\left(\mathcal{S},\mathcal{W}\right). \tag{7}$$

**Corollary 1.** *Given a labeled data point $\left(\mathbf{x}, c\left(\mathbf{x}\right)\right)$ that is correctly classified by an NPC according to Theorem 2 and a corresponding adversarial perturbation $\boldsymbol{\delta}$ that changes the assigned class label.*

*Then, the following inequality is true and has tight bounds (existence of data points for which the equality is true):*

$$\operatorname{margin}_h\left(\{\mathbf{x}\},\mathcal{W}\right) \leq \operatorname{margin}_s\left(\{\mathbf{x}\},\mathcal{W}\right) \leq \|\boldsymbol{\delta}\|. \tag{8}$$

A direct result of this corollary is that we can use the hypothesis margin to reject adversarial examples with perfect recall during inference. Similar to Wong and Kolter [40, Corollary 2]: If a sample $\mathbf{x}$ has a hypothesis margin greater than a certain threshold $\epsilon$, then there exists no *original* sample that can be perturbed with an $\epsilon$-limited adversarial attack such that its classification changes. Thus, $\mathbf{x}$ is certified to be not an adversarial example. In Section 5, we investigate the related false rejection rate.

### 4.3 Hypothesis margin maximization leads to adversarially robust models

We can now define a signed version of the hypothesis margin that incorporates a given class label.

**Definition 4** (signed hypothesis margin). Given a labeled input sample $(\mathbf{x}, c(\mathbf{x}))$ and a set $\mathcal{W}$ of prototypes. The *signed hypothesis margin* of an input sample is

$$\operatorname{margin}_h^c\left(\mathbf{x}, c(\mathbf{x}),\mathcal{W}\right) = \begin{cases} \operatorname{margin}_h\left(\{\mathbf{x}\},\mathcal{W}\right) & \text{if } \mathbf{x} \text{ is correctly classified,} \\ -\operatorname{margin}_h\left(\{\mathbf{x}\},\mathcal{W}\right) & \text{otherwise.} \end{cases} \tag{9}$$

Note that based on Theorem 1, the signed hypothesis margin is lower bounded by the *absolute distance difference* $\triangle(\mathbf{x})$:

$$\triangle(\mathbf{x}) = \left\|\mathbf{x} - \mathbf{w}^-\right\| - \left\|\mathbf{x} - \mathbf{w}^+\right\| \leq 2 \cdot \operatorname{margin}_h^c\left(\mathbf{x}, c(\mathbf{x}),\mathcal{W}\right), \tag{10}$$

where the equality holds if $\mathbf{w}^+$ is equal to $\mathbf{w}^*$ or $\mathbf{w}_*$ (see Equation (2) for the definition of $\mathbf{w}^+$ and $\mathbf{w}^-$). Given a labeled test dataset $\mathcal{T}$ and a set $\mathcal{W}$ of prototypes—based on Equation (10)—we can calculate an upper bound on the robust test error under $\epsilon$-limited adversarial attacks by

$$\operatorname{error}_\epsilon\left(\mathcal{T},\mathcal{W}\right) = \frac{\#\left\{(\mathbf{x}, c(\mathbf{x})) \in \mathcal{T} \mid \left\|\mathbf{x} - \mathbf{w}^-\right\| - \left\|\mathbf{x} - \mathbf{w}^+\right\| \leq 2\epsilon\right\}}{\#\mathcal{T}}. \tag{11}$$

Therefore, by maximizing $\triangle(\mathbf{x})$, we maximize the number of correctly classified samples with a large margin. Moreover, by Corollary 1, we can say that maximizing $\triangle(\mathbf{x})$ maximizes the robustness against adversarial examples or, in other words, minimizing $-\triangle(\mathbf{x})$ maximizes the adversarial robustness. Furthermore, the expression $-\triangle(\mathbf{x})$ is, in fact, a *triplet loss*. Consequently, to optimize an NPC for attacks less than $\epsilon$-limited adversarial attacks one might optimize

$$\frac{1}{\#\mathcal{T}} \sum_{(\mathbf{x}, c(\mathbf{x})) \in \mathcal{T}} \operatorname{ReLU}\left(\left\|\mathbf{x} - \mathbf{w}^+\right\| - \left\|\mathbf{x} - \mathbf{w}^-\right\| + 2\epsilon\right). \tag{12}$$

How does this result apply to the realizations proposed in Section 3? The result above states that as long as we optimize an NPC with a triplet loss, the NPC becomes adversarially robust. However, it is also known that optimizing an NPC with a triplet loss could lead to instabilities in training, which is the reason for the normalization in Equation (2). This normalization impacts the trade-off between large and small margins, but the loss still performs a margin maximization and optimizes for adversarial robustness. Additionally, this result remains true if squared seminorms are used to avoid the possible computation of the square root because the following holds:

$$\left\|\mathbf{x} - \mathbf{w}^+\right\|^2 - \left\|\mathbf{x} - \mathbf{w}^-\right\|^2 = \left(\left\|\mathbf{x} - \mathbf{w}^+\right\| + \left\|\mathbf{x} - \mathbf{w}^-\right\|\right)\left(\left\|\mathbf{x} - \mathbf{w}^+\right\| - \left\|\mathbf{x} - \mathbf{w}^-\right\|\right).$$

Consequently, we can expect that GLVQ and GTLVQ models become adversarially robust during training. In contrast, due to the softmax squashing and the cross-entropy loss, we cannot guarantee that RSLVQ models are robust against adversarial attacks.

## 5 Experiments

In order to verify the presented theoretical results, we performed an experimental analysis on the MNIST [64] and CIFAR-10 [65] datasets. By training several NPCs, we show that the certificate provided by Equation (11) is tight and applies to different $L^p$-norms. We compare the certificate

Table 1: Comparison of NPCs trained with the $L^\infty$-norm against state-of-the-art methods. Dashes "–" indicate that the quantity is not calculable or reported.

| Dataset | Class | Model | CTE [%] | LRTE [%] | URTE [%] |
|---|---|---|---|---|---|
| MNIST $\epsilon = 0.3$ | Certify | GLVQ (128 ppc) | 3.66 | 16.39 | 20.58 |
| | | RSLVQ (128 ppc) | 1.70 | 100.00 | – |
| | | RT [50, Table 3] | 2.68 | 12.46 | 12.46 |
| | | CAP [12, Table 2 "Small"] | 14.87 | – | 43.10 |
| | Verify | RS [39, Table 3 "RS+"] | 2.67 | 7.95 | 19.32 |
| | | IBP [33, Table 4] | 1.66 | 6.12 | 8.05 |
| CIFAR-10 $\epsilon = 8/255$ | Certify | GLVQ (64 ppc) | 59.35 | 79.54 | 79.62 |
| | | RSLVQ (128 ppc) | 54.71 | 99.04 | – |
| | | RT [50, Table 3] | 58.46 | 74.69 | 74.69 |
| | | CAP [12, Table 2 "Resnet"] | 71.33 | – | 78.22 |
| | Verify | RS [39, Table 3 "RS+"] | 59.55 | 73.22 | 79.73 |
| | | IBP [33, Table 4] | 50.51 | 65.23 | 67.96 |

with other methods for guaranteed robustness to highlight that NPCs have comparable guaranteed robustness against adversarial attacks. Particularly, the robust NPCs outperform all other methods in terms of the computational complexity for deriving the certificate. Based on this property, we show that NPCs are the first methods able to perform "real-time" adversarial rejection during inference.

For the $L^\infty$-norm, we compare GLVQ and RSLVQ with ReLU Stability training (RS) [39], Interval Bound Propagation (IBP) [33], Robust Trees (RT) [50], and Convex outer Adversarial Polytope (CAP) [12]. Out of these four, CAP and RT are certification methods. IBP and RS are robustification approaches analyzed by a verification approach.[3] For the $L^2$-norm, we compare GLVQ and GTLVQ with CAP, Stability Training with Noise (STN) [49], and randomized Smoothing (Smooth) [18]. Except for RT, all benchmark methods are based on NNs. Unless otherwise stated, all NPCs are trained by optimizing their respective loss function with stochastic gradient descent, see Section 3. We report the number of prototypes per class (ppc) and the subspace dimension $m$ in the tables.

In Table 1 and Table 2, the results of the comparison are presented. We report the Clean Test Error (CTE) and an Upper bound on the Robust Test Error (URTE). For the $L^\infty$-norm, see Table 1, we also present a Lower bound on the Robust Test Error (LRTE), obtained using the Projected Gradient Descent (PGD) attack [41], and compare NPCs with certification and verification methods. Both LRTE and URTE are evaluated regarding $\epsilon$-limited adversarial attacks—for example, for NPCs, the URTE regarding a certain $\epsilon$ is calculated by Equation (11). The $\epsilon$'s are selected in accordance with reported results in the literature. To compare space and time complexity of the certification methods, we present the number of trainable parameters (#param.) and the number of forward passes[4] (forw. pass.) required to certify an input in the $L^2$-norm setting, see Table 2—all methods presented there are *certification* methods. Further details about the experimental setting, together with additional results, can be found in the supplementary material Section D and Section E. Moreover, besides the evaluation on image datasets, we provide in the supplementary material Section F an evaluation and comparison of NPCs with robust tree-based methods on tabular data. The source code for training and evaluation is available at `https://github.com/saralajew/robust_NPCs`.

**Results of the comparison** As expected, there is a large difference in robustness between the NPCs trained with a triplet loss (GLVQ and GTLVQ) and those trained with a different loss (RSLVQ), see Table 1. In addition to not being able to provide a guarantee, RSLVQ is not empirically robust—as shown by the trivial LRTE. GLVQ and GTLVQ, on the other hand, do provide a nontrivial robustness certificate comparable to, or even better than, the results of an NN trained with CAP as presented in Table 1 and Table 2. In combination with the small gap between LRTE and URTE in Table 1,

Table 2: Comparison of NPCs trained with the $L^2$-norm against state-of-the-art certification methods based on NNs. Values denoted with $^*$ were estimated from figures from the original publication. GTLVQ$^\dagger$ was trained with the loss function from Equation (12) with an $\epsilon$ value of 1.58.

| Dataset | Model | CTE [%] | URTE [%] | Forw. pass. | #param. |
|---|---|---|---|---|---|
| MNIST $\epsilon = 1.58$ | GLVQ (256 ppc) | 4.19 | 65.61 | 1 | 2.0 M |
| | GTLVQ$^\dagger$ (10 ppc, $m = 12$) | 2.92 | 55.32 | 1 | 1.0 M |
| | CAP [12, Table 4 "Large"] | 11.88 | 55.47 | $\geq 749$ | 2.0 M |
| | STN [49, Table 1] | 1.10 | 31.00 | 100 | 0.7 M |
| CIFAR-10 $\epsilon = {}^{36}/_{255}$ | GLVQ (128 ppc) | 51.41 | 61.90 | 1 | 3.9 M |
| | GTLVQ (1 ppc, $m = 100$) | 40.53 | 55.96 | 1 | 3.1 M |
| | CAP [12, Table 2 "Resnet"] | 38.80 | 48.04 | $\geq 3073$ | 4.2 M |
| | STN [49, Table 1] | 19.50 | 34.40 | 100 | 1.4 M |
| | Smooth [18, Figure 6 top, 0.12] | 18$^*$ | 27$^*$ | 100100 | 1.7 M |

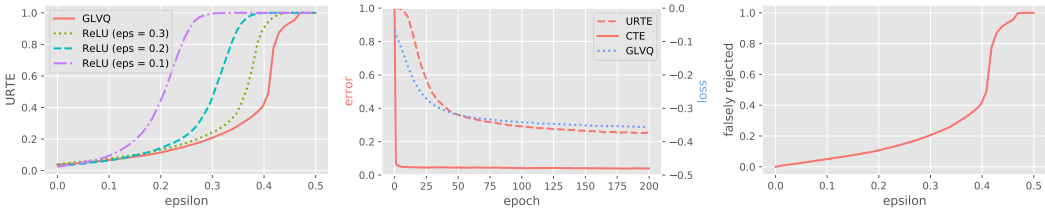

Figure 1: Different results on the MNIST test dataset of GLVQ models trained with the $L^\infty$-norm. Left: URTE after training with different losses. Middle: Evolution of CTE, GLVQ loss, and URTE ($\epsilon = 0.3$) during training. Right: Ratio of falsely rejected samples by the hypothesis margin.

this implies that the bound based on the hypothesis margin is not only formally but also empirically tight. However, the results from the same table show that the robust NPCs do not improve over the other models trained for high robustness but are reasonably close in terms of the URTE. Compared to current state-of-the-art robustness certification methods based on randomized smoothing (STN and Smooth) presented in Table 2, there is a considerable deficit in terms of URTE. However, it must be noted that these methods are not deterministic and require extensive sampling to compute the certificate (and prediction). To illustrate, Smooth [18, Section 4] takes *15 seconds to certify a sample from CIFAR-10*, while GLVQ *certifies the entire test dataset during this time*. This difference is also depicted in Table 2 by the number of forward passes needed to certify a single input.

**Some notes on training adversarially robust NPCs**   We considered two triplet losses for training GLVQ and GTLVQ: the ReLU clipped absolute distance difference loss of Equation (12) and the GLVQ loss of Equation (2). While the ReLU loss optimizes for predetermined $\epsilon$ in $\epsilon$-limited attacks, the GLVQ loss does not. Despite the former being more similar to other certification methods, we found that using the GLVQ loss often results in the highest guaranteed robustness for almost all $\epsilon$'s, as shown in Figure 1 (left). Second, all NPCs are trained without early stopping based on plateauing CTE. Optimizing a triplet loss is directly tied to maximizing the guaranteed robustness. Hence, as long as the triplet loss is improving, the guarantee improves too, see Figure 1 (middle). This is an argument against early stopping based on plateauing CTE. Crucially, the URTE can be logged during training as it only requires one extra floating-point operation per sample.

**Real-time adversarial rejection**   As stated in Section 4.2, the hypothesis margin can be used to reject adversarial examples with perfect recall. While this view on guaranteed adversarial robustness was voiced before, it suffered from the computational overhead of certifying a sample. With the hypothesis margin calculable with one floating-point operation after inference, it does not suffer from the same problem. To evaluate the rejection strategy, we apply it to the MNIST test dataset and the GLVQ model trained with the $L^\infty$-norm. The false positive rate for different values of $\epsilon$ is given in Figure 1 (right). Note that the adversarial rejection strategy is guaranteed to provide perfect recall, hence, investigating only the falsely rejected sample suffices. We find that with an $\epsilon$ of 0.3, the

adversarial rejection strategy falsely rejects only 20 % of all samples. To emphasize, this is achieved without excessive overhead and perfect recall. Further investigation of the falsely rejected samples in supplementary Section E.3 shows that they are semantically close to the hypothetical original class. For example, most rejected samples from the class 9 are close to a prototype of the class 4.

# 6    Discussion

The experimental evaluation presents us with the following results: First of all, training NPCs with a triplet loss is an effective robustification strategy. The resulting models are empirically robust and their guaranteed robustness is tight, nontrivial, improves over other certification methods, and yields comparable performance even when compared with verification methods. More importantly, we showed a large difference in computational overhead to derive the guarantees between NN certifiers (verifiers) and the NPC certification.

There are also downsides to the work presented that we would like to clarify here. First, the method does not improve over state of the art and we expect issues in scaling to high dimensional datasets like ImageNet [66] without a proper feature engineering—this is a common challenge in NPCs. However, if resources and time are scarce, none of the current state-of-the-art methods are applicable. Hence, we do not consider not improving over these methods as a critical limitation. A second problem is that, in principle, it would be desired to guarantee robustness for several seminorms with one NPC. Unfortunately, the presented certificates are always related to the seminorm used to classify the data. But if we train a robust NPC with a *fixed $L^p$-norm*, Hölder's inequality provides a simple bound on the size of adversarial perturbations $\boldsymbol{\delta}$ regarding other $L^q$-norms: $\|\boldsymbol{\delta}\|_q \geq \mathrm{margin}_h\left(\{\mathbf{x}\}, \mathcal{W}\right)$ if $q \leq p$ and $\|\boldsymbol{\delta}\|_q \geq n^{\frac{1}{q}-\frac{1}{p}} \mathrm{margin}_h\left(\{\mathbf{x}\}, \mathcal{W}\right)$ otherwise. Preliminary results regarding this can be found in the supplementary material Section E.2.

We close this section with some concluding remarks considering the importance of the generality of the results as they hold for an arbitrary seminorm. A common subset of NPCs are those that use an adaptive dissimilarity measure—for example, GMLVQ and LMNN [19] with a 1-nearest neighbor rule. Both methods use the same dissimilarity measure: the quadratic form $d\left(\mathbf{x}, \mathbf{w}\right) = \|\mathbf{Q}\left(\mathbf{x} - \mathbf{w}\right)\|_2$ with $\mathbf{Q} \in \mathbb{R}^{m \times n}$. The matrix $\mathbf{Q}$ can be considered to encode the feature relevance with respect to the classification task, which is an important characteristic—for example, it is used to identify relevant biomarkers for malignancy of adrenal tumors [67, 68]. Based on the presented theory, both methods are hypothesis margin maximizers regarding the seminorm $\|\mathbf{x}\|_{\mathbf{Q}} = \|\mathbf{Q}\mathbf{x}\|_2$ since both optimize a triplet loss. Therefore, we can expect strong guaranteed robustness with respect to attacks that optimize $\|\boldsymbol{\delta}\|_{\mathbf{Q}}$. However, being robust regarding $\|\boldsymbol{\delta}\|_{\mathbf{Q}}$ optimized attacks does not necessarily imply robustness regarding $L^p$-norms. Hence, in commonly used empirical robustness evaluation settings, adaptive measure NPCs might seem to be non-robust—as empirically observed for GMLVQ [26].

# 7    Conclusion and outlook

In this work, we presented a theory to robustify and certify NPCs with respect to an *arbitrary* seminorm. To the best of our knowledge, this is the most universal and practically applicable approach with nontrivial robustness guarantees. Based on the hypothesis margin, we have proven an efficiently calculable and tight lower bound on the robust test error of an NPC. The numerical evaluation of this bound showed that the robustness guarantee of NPCs surpassed other NN-based certification methods and is close to verification methods. At the same time, it significantly improved the computational complexity. Together with their inherent interpretability [20, 22], NPCs are a great alternative for NNs in the adversarial setting and the superior choice when compute time is restricted.

To improve the presented results, we suggest the study of NPCs in the context of ensembles (similar to RTs [50]) and cascade models [12]. In the previous section, we discussed how robustness guarantees can be computed for various $L^p$-norms simultaneously. Since these guarantees are often too weak in practice, future work should examine whether the results of Croce and Hein [13] can be generalized to NPCs. Additionally, further studies should cover whether the idea of the hypothesis margin—*varying parameters instead of inputs* to derive a calculable margin—can be extended to NNs.

## Broader impact

With the more widespread application of machine learning methods in our everyday life, the potential negative impact of adversarial attacks on society increases. As discussed in the introduction, neither current empirical robustness methods nor certification or verification methods are sufficient to eliminate this problem. For applied machine learning research in medium to large companies, the current state-of-the-art methods for certifying or verifying adversarial robustness require a too large investment in compute time to truly incorporate the guaranteed robustness of a model as a formal requirement for the productization of machine learning. The theoretical robustness bound presented in this work can however be parallelized with the accuracy evaluation of a model and can therefore be easily incorporated in the already existing evaluation pipelines. With the upper bound on the robust test error calculable in constant time, it is even possible to incorporate the certification of an NPC as a metric in the training procedure—outputting the certified adversarial robustness after each epoch. A potential application of the reduced impact on inference time is also discussed in Section 5.

Although deep neural networks frequently deliver excellent performances, the interpretability of those networks is difficult [69]. Recently, this has led to a wide line of research into the development of interpretable models, particularly for technical and medical applications [23, 24, 70, 71]. Having this in mind, the consideration of NPCs, as one of the most prominent interpretable paradigms [22], is of general interest. However, to ensure that interpretable models can be used without unwanted negative side effects, it is important to investigate their properties to the same extent as has been done for non-interpretable models. The investigation of the guaranteed adversarial robustness of NPCs is, therefore, a crucial step in this transition. In addition to this, the positive definiteness of norm-based distances can impose a significant restriction on the dissimilarity measure used in NPCs. By showing that this is not a requirement for constructing an adversarially robust NPC, this restriction is removed. Hence, more freedom is obtained in the selection of dissimilarity measures—for example, adaptive dissimilarity measures, as discussed in Section 6. This allows the application of NPCs as interpretable models in a wider variety of use cases.

To summarize, we foresee two potential areas where the theoretical work presented here could have a direct and lasting impact on society. First, with the upper bound on the robust test error calculable in constant-time, the certification method presented here is more suitable for direct incorporation in the development of machine learning methods. This has been extensively discussed and evaluated in previous sections. Second, as a side effect, with NPCs now proven to be robust against adversarial attacks, they are better suited and more widely applicable as an interpretable alternative to NNs in real-world applications.

## Acknowledgments and disclosure of funding

We would like to thank Peter Schlicht for his valuable contribution to earlier versions of the manuscript and Eric Wong for his helpful discussion about CAP. Moreover, we would like to thank our attentive anonymous AC and reviewers whose comments have greatly improved this manuscript.

None of the authors received third party funding or have had any financial relationship with entities that could potentially be perceived to influence the submitted work during the 36 months prior to this submission.

## Footnotes

[2]Compared to the other NPC methods, RSLVQ is a *probabilistic* version of a prototype-based classifier and *not* an NPC according to our definition. However, for simplicity, we consider RSLVQ as NPC in the following comparisons because it uses prototypes with fixed class labels and a dissimilarity. But note that it classifies and trains according to a probabilistic approach.

[3]For clarification, *IBP also provides a certificate* (see Table 3 in the referenced publication), but we refer to the robustness results achieved by a verification approach to compare with strong adversarial robustness results—note Footnote 4 in the referenced publication regarding possible overestimation of the reported errors.

[4]The CAP certificate is calculated using a dual network. Based on a discussion with the authors, we found that the number of forward passes in the full network can be approximated by the presented lower bounds.

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
