[Supplementary Material]

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

Figure 2: Visualization of the defined margins for a GLVQ model with the Euclidean distance in $\mathbb{R}^2$. The softly colored dots are the training samples from the class "orange" or "blue". The fully colored dot is a new sample that has to be classified by the model. The two stars represent the prototypes and thus the model weights trained by the GLVQ algorithm. The solid black line is the resulting decision boundary. Left: The final weight configuration of the trained GLVQ model including the margins (dashed circles) with respect to the given data sample. Right: The perturbed weight configuration that changes the assigned class label of the new data sample. Note that the prototypes have been shifted by a magnitude equal to the hypothesis margin.

## A   Visualization of the hypothesis and sample margins

In Figure 2, we visualize the effect of changing the weights of a GLVQ model by a magnitude equal to the hypothesis margin (with respect to a given sample). We note that if we span a ball with a radius equal to the hypothesis margin around the prototypes, the prototypes can be placed at an arbitrary position *inside* these balls without changing the assigned class label of the sample. Likewise, as the right visualization shows, we can systematically find a position for the prototypes such that they have been shifted by a magnitude *equal* to the hypothesis margin such that the prototypes assign a different class label to the given sample.

In the next sections, we use the observation that the prototypes have been shifted directly towards or away from the data point to formally prove the calculation of the hypothesis margin and the relation between the hypothesis and sample margin.

## B   Proof of Theorem 1: Calculation of the hypothesis margin

**Theorem.** *Let the data space $\mathcal{X}$ be a vector space over a field of the real or complex numbers, $d(\mathbf{x}, \mathbf{w}) = \|\mathbf{x} - \mathbf{w}\|$ be a dissimilarity induced by a seminorm $\|\cdot\|$, and $\mathbf{x} \in \mathcal{X}$ be an input. Then, the hypothesis margin of the set $\mathcal{W}$ of prototypes with respect to $\mathbf{x}$ can be computed by*

$$\text{margin}_h\left(\{\mathbf{x}\}, \mathcal{W}\right) = \frac{1}{2}\left(\|\mathbf{x} - \mathbf{w}_*\| - \|\mathbf{x} - \mathbf{w}^*\|\right), \tag{13}$$

*where $\mathbf{w}^*$ denotes the closest prototype to $\mathbf{x}$ and $\mathbf{w}_*$ denotes the closest prototype to $\mathbf{x}$ with a different class label than the class label of $\mathbf{w}^*$.*

The proof is based on the ideas used by Crammer et al. [56].

*Proof.* The outline of the proof is as follows:

1. Given a set $\mathcal{W}$ of prototypes, we define a set $\hat{\mathcal{W}}$ of shifted prototypes in which the prototypes are shifted to an arbitrary position within the corresponding balls of radius

$$r = \frac{1}{2}\left(\|\mathbf{x} - \mathbf{w}_*\| - \|\mathbf{x} - \mathbf{w}^*\|\right). \tag{14}$$

Using this set of shifted prototypes, we prove that

$$\text{margin}_h\left(\{\mathbf{x}\}, \mathcal{W}\right) \geq r.$$

2. For an arbitrary but sufficiently small $\varepsilon > 0$, we define a *second* set $\hat{\mathcal{W}}$ of shifted prototypes in which the prototypes are shifted by a vector of length $r + \frac{\varepsilon}{2}$. Based on this set, we prove that it assigns a different class label to the input $\mathbf{x}$. Consequently, we can conclude that

$$\text{margin}_h\left(\{\mathbf{x}\}, \mathcal{W}\right) \leq r.$$

Both steps together prove the theorem.

We use the following notations in the proof: Let $\mathbf{x}$, $\mathbf{w}^*$, and $\mathbf{w}_*$ be defined as in the theorem, $\mathbf{w}$ be an arbitrary prototype of $\mathcal{W}$, $\mathbf{w}_\diamond$ be an arbitrary prototype of $\mathcal{W}$ with a class label different than $c\left(\mathbf{w}^*\right)$, and $r$ be defined as in Equation (14). With the dissimilarity induced by a seminorm, we use the notations $d\left(\mathbf{x}, \mathbf{w}\right)$ and $\|\mathbf{x} - \mathbf{w}\|$ interchangeably.

**Step 1:** Let $\hat{\mathcal{W}}$ be a set of shifted prototypes constructed by repositioning each prototype in $\mathcal{W}$ to an arbitrary position within its induced ball of radius $r$. Formally, this is realized by defining an arbitrary function $\mathbf{s} : \mathcal{W} \to \mathcal{X}$ such that $\|\mathbf{s}\left(\mathbf{w}\right)\| < r$. The shifted prototype to $\mathbf{w}$ is obtained by

$$\hat{\mathbf{w}}\left(\mathbf{w}\right) = \mathbf{w} + \mathbf{s}\left(\mathbf{w}\right).$$

All shifted prototypes combined, provide the set $\hat{\mathcal{W}}$ of shifted prototypes.

Given an arbitrary prototype $\mathbf{w}$, we conclude that the dissimilarity between the shifted and the non-shifted prototype is always less than $r$:

$$d\left(\mathbf{w}, \hat{\mathbf{w}}\left(\mathbf{w}\right)\right) = \|\mathbf{w} - \hat{\mathbf{w}}\left(\mathbf{w}\right)\| = \|\mathbf{s}\left(\mathbf{w}\right)\| < r.$$

Using the triangle inequality, we can state that the dissimilarity between a shifted prototype and the input $\mathbf{x}$ is less than $d\left(\mathbf{x}, \mathbf{w}\right) + r$:

$$d\left(\mathbf{x}, \hat{\mathbf{w}}\left(\mathbf{w}\right)\right) \leq d\left(\mathbf{x}, \mathbf{w}\right) + d\left(\mathbf{w}, \hat{\mathbf{w}}\left(\mathbf{w}\right)\right),$$
$$< d\left(\mathbf{x}, \mathbf{w}\right) + r. \tag{15}$$

Similarly, using the triangle inequality again, we can state that

$$d\left(\mathbf{x}, \hat{\mathbf{w}}\left(\mathbf{w}\right)\right) \geq d\left(\mathbf{x}, \mathbf{w}\right) - d\left(\mathbf{w}, \hat{\mathbf{w}}\left(\mathbf{w}\right)\right),$$
$$> d\left(\mathbf{x}, \mathbf{w}\right) - r. \tag{16}$$

Given a prototype $\mathbf{w}_\diamond \in \mathcal{W}$ with a different class label than $c\left(\mathbf{w}^*\right)$, it follows that

$$d\left(\mathbf{x}, \mathbf{w}_\diamond\right) \geq d\left(\mathbf{x}, \mathbf{w}_*\right).$$

Using Equation (16), we conclude

$$d\left(\mathbf{x}, \hat{\mathbf{w}}\left(\mathbf{w}_\diamond\right)\right) > d\left(\mathbf{x}, \mathbf{w}_\diamond\right) - r \geq d\left(\mathbf{x}, \mathbf{w}_*\right) - r. \tag{17}$$

By the definition of $r$, see Equation (14), it holds that

$$d\left(\mathbf{x}, \mathbf{w}^*\right) + r = d\left(\mathbf{x}, \mathbf{w}^*\right) + \frac{1}{2}\left(d\left(\mathbf{x}, \mathbf{w}_*\right) - d\left(\mathbf{x}, \mathbf{w}^*\right)\right),$$
$$= d\left(\mathbf{x}, \mathbf{w}_*\right) - r. \tag{18}$$

Now, combining Equation (15) for the shifted prototype $\hat{\mathbf{w}}\left(\mathbf{w}^*\right)$ with Equation (18) and Equation (17), we obtain

$$d\left(\mathbf{x}, \hat{\mathbf{w}}\left(\mathbf{w}^*\right)\right) < d\left(\mathbf{x}, \mathbf{w}^*\right) + r = d\left(\mathbf{x}, \mathbf{w}_*\right) - r < d\left(\mathbf{x}, \hat{\mathbf{w}}\left(\mathbf{w}_\diamond\right)\right).$$

With $\mathbf{w}_\diamond$ being an arbitrary prototype of $\mathcal{W}$ with a class label different than $c\left(\mathbf{w}^*\right)$, this states that each shifted prototype $\hat{\mathbf{w}}\left(\mathbf{w}_\diamond\right)$ has a larger dissimilarity than $\hat{\mathbf{w}}\left(\mathbf{w}^*\right)$ to the input sample $\mathbf{x}$. In other words, the closest shifted prototype in $\hat{\mathcal{W}}$ to $\mathbf{x}$ has the same class label as the closest prototype in $\mathcal{W}$—in symbols, $c_{\hat{\mathcal{W}}}^*\left(\mathbf{x}\right) = c_{\mathcal{W}}^*\left(\mathbf{x}\right)$. Since the shift of the prototypes of $\mathcal{W}$ to the new positions in $\hat{\mathcal{W}}$ was arbitrary, we conclude that *the hypothesis margin of $\mathcal{W}$ is greater than or equal to $r$*.

**Step 2:** For an arbitrary $\varepsilon > 0$ that is less than or equal to $\|\mathbf{x} - \mathbf{w}_*\|$, we define a *new* set $\hat{\mathcal{W}}$ of shifted prototypes of $\mathcal{W}$ by shifting each prototype by a magnitude of $r + \frac{\varepsilon}{2}$. Therefore, we relocate the prototypes inside balls of size $r + \varepsilon$. For each prototype $\mathbf{w} \in \mathcal{W}$, we define a unit vector with respect to $\|\cdot\|$ by

$$
\mathbf{z}\,(\mathbf{w}) = \begin{cases} \mathbf{u} & \text{if } \|\mathbf{x} - \mathbf{w}\| = 0, \\ \frac{\mathbf{x} - \mathbf{w}}{\|\mathbf{x} - \mathbf{w}\|} & \text{otherwise,} \end{cases}
\tag{19}
$$

where $\mathbf{u}$ is an arbitrary unit vector. Because the set $\mathcal{W}$ of prototypes assigns a class label unambiguously to $\mathbf{x}$, the seminorm in Equation (19) vanishes only for prototypes of the class $c\,(\mathbf{w}^*)$.

The set $\hat{\mathcal{W}}$ of shifted prototypes is defined by the shifted prototypes $\hat{\mathbf{w}}\,(\mathbf{w})$ according to the following equation:

$$
\hat{\mathbf{w}}\,(\mathbf{w}) = \begin{cases} \mathbf{w} + \left(r + \frac{\varepsilon}{2}\right) \mathbf{z}\,(\mathbf{w}) & \text{if } c\,(\mathbf{w}) \neq c\,(\mathbf{w}^*), \\ \mathbf{w} - \left(r + \frac{\varepsilon}{2}\right) \mathbf{z}\,(\mathbf{w}) & \text{otherwise.} \end{cases}
$$

Keeping in mind the absolute homogeneity of a seminorm, we can state that for each prototype $\hat{\mathbf{w}}\,(\mathbf{w}) \in \hat{\mathcal{W}}$ that has the *same* class label as $\mathbf{w}^*$, the equality

$$
\begin{aligned}
\|\mathbf{x} - \hat{\mathbf{w}}\,(\mathbf{w})\| &= \left\| (\mathbf{x} - \mathbf{w}) \left( 1 + \frac{r + \frac{\varepsilon}{2}}{\|\mathbf{x} - \mathbf{w}\|} \right) \right\|, \\
&= \left( 1 + \frac{r + \frac{\varepsilon}{2}}{\|\mathbf{x} - \mathbf{w}\|} \right) \|\mathbf{x} - \mathbf{w}\|, \\
&= \|\mathbf{x} - \mathbf{w}\| + r + \frac{\varepsilon}{2}
\end{aligned}
\tag{20}
$$

holds. Note that this equation is also valid if $\|\mathbf{x} - \mathbf{w}\| = 0$: From the triangle inequality, it follows that

$$
\begin{aligned}
\|\mathbf{x} - \hat{\mathbf{w}}\,(\mathbf{w})\| &= \left\| (\mathbf{x} - \mathbf{w}) + \left( r + \frac{\varepsilon}{2} \right) \mathbf{u} \right\|, \\
&\leq \|\mathbf{x} - \mathbf{w}\| + \left\| \left( r + \frac{\varepsilon}{2} \right) \mathbf{u} \right\|, \\
&\leq r + \frac{\varepsilon}{2}.
\end{aligned}
\tag{21}
$$

Similarly, the triangle inequality implies that the seminorm of $\left( r + \frac{\varepsilon}{2} \right) \mathbf{u}$ is bounded by $\|\mathbf{x} - \hat{\mathbf{w}}\,(\mathbf{w})\|$:

$$
\begin{aligned}
\left\| \left( r + \frac{\varepsilon}{2} \right) \mathbf{u} \right\| &= \left\| \left( r + \frac{\varepsilon}{2} \right) \mathbf{u} + (\mathbf{x} - \mathbf{w}) - (\mathbf{x} - \mathbf{w}) \right\|, \\
&\leq \left\| (\mathbf{x} - \mathbf{w}) + \left( r + \frac{\varepsilon}{2} \right) \mathbf{u} \right\| + \|\mathbf{x} - \mathbf{w}\|, \\
&\leq \|\mathbf{x} - \hat{\mathbf{w}}\,(\mathbf{w})\|.
\end{aligned}
$$

Therefore, we obtain

$$
\|\mathbf{x} - \hat{\mathbf{w}}\,(\mathbf{w})\| \geq r + \frac{\varepsilon}{2}.
\tag{22}
$$

The combination of Equation (21) and Equation (22) yields equality:

$$
\|\mathbf{x} - \hat{\mathbf{w}}\,(\mathbf{w})\| = r + \frac{\varepsilon}{2}.
$$

Consequently, Equation (20) is valid for *all* prototypes $\hat{\mathbf{w}}\,(\mathbf{w})$ with the *same* class label as $\mathbf{w}^*$.

Analogously, if $\hat{\mathbf{w}}\,(\mathbf{w}) \in \hat{\mathcal{W}}$ has a *different* class label than $c\,(\mathbf{w}^*)$, the seminorm of $\mathbf{x} - \hat{\mathbf{w}}\,(\mathbf{w})$ becomes

$$
\begin{aligned}
\|\mathbf{x} - \hat{\mathbf{w}}\,(\mathbf{w})\| &= \left\| (\mathbf{x} - \mathbf{w}) \left( 1 - \frac{r + \frac{\varepsilon}{2}}{\|\mathbf{x} - \mathbf{w}\|} \right) \right\|, \\
&= \left| 1 - \frac{r + \frac{\varepsilon}{2}}{\|\mathbf{x} - \mathbf{w}\|} \right| \|(\mathbf{x} - \mathbf{w})\|, \\
&= \left| \|\mathbf{x} - \mathbf{w}\| - r - \frac{\varepsilon}{2} \right|.
\end{aligned}
\tag{23}
$$

With $\mathbf{w}_*$ being the closest prototype with a class label other than $c(\mathbf{w}^*)$ and $\varepsilon$ less than or equal to $\|\mathbf{x} - \mathbf{w}_*\|$, it follows that

$$\|\mathbf{x} - \mathbf{w}_\diamond\| - r - \frac{\varepsilon}{2} \geq \|\mathbf{x} - \mathbf{w}_\diamond\| - \frac{1}{2}\left(\|\mathbf{x} - \mathbf{w}_*\| - \|\mathbf{x} - \mathbf{w}^*\|\right) - \frac{1}{2}\|\mathbf{x} - \mathbf{w}_*\|,$$

$$\geq \|\mathbf{x} - \mathbf{w}_\diamond\| - \|\mathbf{x} - \mathbf{w}_*\| + \frac{1}{2}\|\mathbf{x} - \mathbf{w}^*\|,$$

and, hence, we obtain

$$\|\mathbf{x} - \mathbf{w}_\diamond\| - r - \frac{\varepsilon}{2} \geq 0.$$

This implies that the argument of the absolute value function of Equation (23) is always positive so that

$$\|\mathbf{x} - \hat{\mathbf{w}}(\mathbf{w})\| = \|\mathbf{x} - \mathbf{w}\| - r - \frac{\varepsilon}{2} \tag{24}$$

for all prototypes with a different class label than $c(\mathbf{w}^*)$.

Using Equation (24) for the shifted prototype $\hat{\mathbf{w}}(\mathbf{w}_*)$, we get

$$\|\mathbf{x} - \hat{\mathbf{w}}(\mathbf{w}_*)\| = \|\mathbf{x} - \mathbf{w}_*\| - \frac{1}{2}\left(\|\mathbf{x} - \mathbf{w}_*\| - \|\mathbf{x} - \mathbf{w}^*\|\right) - \frac{\varepsilon}{2},$$

$$= \frac{1}{2}\left(\|\mathbf{x} - \mathbf{w}_*\| + \|\mathbf{x} - \mathbf{w}^*\|\right) - \frac{\varepsilon}{2}. \tag{25}$$

Similarly, using Equation (20) for the shifted prototype $\hat{\mathbf{w}}(\mathbf{w}^*)$, we obtain

$$\|\mathbf{x} - \hat{\mathbf{w}}(\mathbf{w}^*)\| = \|\mathbf{x} - \mathbf{w}^*\| + \frac{1}{2}\left(\|\mathbf{x} - \mathbf{w}_*\| - \|\mathbf{x} - \mathbf{w}^*\|\right) + \frac{\varepsilon}{2},$$

$$= \frac{1}{2}\left(\|\mathbf{x} - \mathbf{w}_*\| + \|\mathbf{x} - \mathbf{w}^*\|\right) + \frac{\varepsilon}{2}. \tag{26}$$

Comparing Equation (25) with Equation (26) and with $\varepsilon$ being positive, we conclude that

$$\|\mathbf{x} - \hat{\mathbf{w}}(\mathbf{w}_*)\| < \|\mathbf{x} - \hat{\mathbf{w}}(\mathbf{w}^*)\|$$

holds. Since this implies that $\hat{\mathbf{w}}(\mathbf{w}_*)$ is the *closest* prototype in the set $\hat{\mathcal{W}}$, the set $\hat{\mathcal{W}}$ of shifted prototypes assigns the class label of $c(\mathbf{w}_*)$ to $\mathbf{x}$ and thus labels $\mathbf{x}$ other than $\mathcal{W}$—in symbols, $c_{\hat{\mathcal{W}}}^*(\mathbf{x}) \neq c_{\mathcal{W}}^*(\mathbf{x})$. Therefore, the hypothesis margin must be less than $r + \varepsilon$ because we constructed $\hat{\mathcal{W}}$ by means of prototype shifts with a magnitude of size $r + \frac{\varepsilon}{2}$. With $\varepsilon > 0$ arbitrarily chosen, it follows that *the hypothesis margin of $\mathcal{W}$ is less than or equal to $r$.*

Now, combining the final result of Step 1 with the final result of Step 2, we obtain that the hypothesis margin is *exactly* $r$. □

## C  Proof of Theorem 2: Relation to the sample margin

**Theorem.** *Let the data space $\mathcal{X}$ be a vector space over a field of the real or complex numbers, $d(\mathbf{x}, \mathbf{w}) = \|\mathbf{x} - \mathbf{w}\|$ be a dissimilarity induced by a seminorm $\|\cdot\|$, and $\mathcal{S}$ be a set of inputs. Then, the hypothesis margin of $\mathcal{W}$ with respect to $\mathcal{S}$ yields a lower bound on the sample margin of $\mathcal{W}$ with respect to $\mathcal{S}$:*

$$\operatorname{margin}_h(\mathcal{S}, \mathcal{W}) \leq \operatorname{margin}_s(\mathcal{S}, \mathcal{W}).$$

The proof is based on the ideas used by Crammer et al. [56].

*Proof.* We prove the theorem by the following steps:

1. Given a set $\mathcal{W}$ of prototypes and an arbitrary radius $r$ that fulfills the requirement

$$\operatorname{margin}_s(\mathcal{S}, \mathcal{W}) < r, \tag{27}$$

   we define a set $\hat{\mathcal{W}}$ of shifted prototypes such that each prototype is shifted by a vector of length $r$.

2. We prove that this set $\hat{\mathcal{W}}$ of prototypes labels the inputs of $\mathcal{S}$ differently than $\mathcal{W}$ so that the hypothesis margin must be less than or equal to $r$. Because the radius $r$ was arbitrarily chosen, it follows that the hypothesis margin must be less than or equal to the sample margin, which proves the theorem.

**Step 1:** Let $r$ be an arbitrary radius such that Equation (27) holds. Because the sample margin $\text{margin}_s(\mathcal{S}, \mathcal{W})$ is less than $r$, there exists an element $\mathbf{x}$ of $\mathcal{S}$ such that there exists another element $\bar{\mathbf{x}}$ of $\mathcal{X}$ with the distance

$$d(\mathbf{x}, \bar{\mathbf{x}}) = \|\mathbf{x} - \bar{\mathbf{x}}\| = r \tag{28}$$

that has an assigned class label different than the class label assigned to $\mathbf{x}$—in symbols, $c^*_{\mathcal{W}}(\mathbf{x}) \neq c^*_{\mathcal{W}}(\bar{\mathbf{x}})$. Because $\mathbf{x}$ and $\bar{\mathbf{x}}$ are labeled differently, the closest prototypes must have different class labels too—we denote by $\mathbf{w}^*$ the closest prototype to $\mathbf{x}$ and by $\bar{\mathbf{w}}^*$ the closest prototype to $\bar{\mathbf{x}}$.

Based on $\mathbf{x}$ and $\bar{\mathbf{x}}$, we define the set $\hat{\mathcal{W}}$ of shifted prototypes of $\mathcal{W}$ by shifting each prototype $\mathbf{w} \in \mathcal{W}$ to the position

$$\hat{\mathbf{w}}(\mathbf{w}) = \mathbf{w} + \mathbf{x} - \bar{\mathbf{x}}. \tag{29}$$

Due to Equation (28), the magnitude of the applied shift to a prototype is exactly $r$:

$$d(\mathbf{w}, \hat{\mathbf{w}}(\mathbf{w})) = \|\mathbf{w} - \mathbf{w} - \mathbf{x} + \bar{\mathbf{x}}\| = r. \tag{30}$$

**Step 2:** By Equation (29), it follows that $d(\bar{\mathbf{x}}, \mathbf{w})$ equals $d(\mathbf{x}, \hat{\mathbf{w}}(\mathbf{w}))$:

$$d(\bar{\mathbf{x}}, \mathbf{w}) = \|\bar{\mathbf{x}} \underbrace{-\mathbf{x} + \mathbf{x}}_{=\mathbf{0}} - \mathbf{w}\| = \|\mathbf{x} - (\mathbf{w} + \mathbf{x} - \bar{\mathbf{x}})\| = d(\mathbf{x}, \hat{\mathbf{w}}(\mathbf{w})). \tag{31}$$

Because $\bar{\mathbf{w}}^*$ is the closest prototype to $\bar{\mathbf{x}}$, the dissimilarity of $\bar{\mathbf{x}}$ to an arbitrary prototype $\mathbf{w}_\diamond \in \mathcal{W}$ of a class *other* than $c(\bar{\mathbf{w}}^*)$ is larger:

$$d(\bar{\mathbf{x}}, \bar{\mathbf{w}}^*) < d(\bar{\mathbf{x}}, \mathbf{w}_\diamond). \tag{32}$$

Combining Equation (32) and Equation (31), we conclude that

$$d(\bar{\mathbf{x}}, \bar{\mathbf{w}}^*) = d(\mathbf{x}, \hat{\mathbf{w}}(\bar{\mathbf{w}}^*)) < d(\mathbf{x}, \hat{\mathbf{w}}(\mathbf{w}_\diamond)) = d(\bar{\mathbf{x}}, \mathbf{w}_\diamond),$$

which implies that $\hat{\mathbf{w}}(\bar{\mathbf{w}}^*)$ must be the closest prototype to $\mathbf{x}$ regarding the set $\hat{\mathcal{W}}$ of prototypes. Therefore, the set $\hat{\mathcal{W}}$ of prototypes assigns a *different* class label than $\mathcal{W}$ to $\mathbf{x}$—in symbols, $c^*_{\hat{\mathcal{W}}}(\mathbf{x}) \neq c^*_{\mathcal{W}}(\mathbf{x})$. Because all prototypes $\mathbf{w} \in \mathcal{W}$ have been shifted by the magnitude $r$, see Equation (30), the hypothesis margin must be less than or equal to $r$. Additionally, since this result holds for an arbitrary $r$ according to Equation (27), the hypothesis margin must be less than or equal to the sample margin. $\qquad\square$

**Corollary.** *Given a labeled data point $(\mathbf{x}, c(\mathbf{x}))$ that is correctly classified by an NPC according to Theorem 2 and a corresponding adversarial perturbation $\boldsymbol{\delta}$ that changes the assigned class label. Then, the following inequality is true and has tight bounds:*

$$\text{margin}_h(\{\mathbf{x}\}, \mathcal{W}) \leq \text{margin}_s(\{\mathbf{x}\}, \mathcal{W}) \leq \|\boldsymbol{\delta}\|.$$

*Proof.* The inequality immediately follows from Theorem 2, Definition 1 of the sample margin, and the requirement that an adversarial perturbation changes the assigned class label. Moreover, by Definition 1 of the sample margin, the closest adversarial example is an element at a distance $\text{margin}_s(\{\mathbf{x}\}, \mathcal{W})$ to $\mathbf{x}$. Therefore, the minimum of Equation (6) is exactly $\text{margin}_s(\{\mathbf{x}\}, \mathcal{W})$—which implies the tightness of the first lower bound. The tightness of the lower bound

$$\text{margin}_h(\{\mathbf{x}\}, \mathcal{W}) \leq \text{margin}_s(\{\mathbf{x}\}, \mathcal{W})$$

is proven by showing that there exists an element $\mathbf{x} \in \mathcal{X}$ such that the hypothesis margin is greater than or equal to the sample margin. Together with Equation (7) of the theorem, this implies equality and the tightness of the bound.

If $\mathbf{x}$ is set to be equal to an arbitrary prototype $\mathbf{w} \in \mathcal{W}$, then the closest prototype $\mathbf{w}^*$ to $\mathbf{x}$ is $\mathbf{w}$. Additionally, we denote by $\mathbf{w}_*$ the closest prototype with a different class label than $c(\mathbf{w}^*)$. The decision boundary between these two prototypes is defined by all elements $\mathbf{x}' \in \mathcal{X}$ that fulfill the following criterion:

$$\|\mathbf{x}' - \mathbf{w}^*\| = \|\mathbf{x}' - \mathbf{w}_*\|.$$

The vector $\bar{\mathbf{x}}$ defined as

$$\begin{aligned} \bar{\mathbf{x}} &= \mathbf{x} + \frac{1}{2}(\mathbf{w}_* - \mathbf{w}^*), \\ &= \mathbf{w}^* + \frac{1}{2}(\mathbf{w}_* - \mathbf{w}^*), \\ &= \frac{1}{2}(\mathbf{w}_* + \mathbf{w}^*) \end{aligned} \tag{33}$$

satisfies this decision-boundary criterion. Hence, the sample margin must be less than or equal to the dissimilarity from $\mathbf{x}$ to the element $\bar{\mathbf{x}}$ from Equation (33):

$$\text{margin}_s\left(\{\mathbf{x}\}, \mathcal{W}\right) \leq d\left(\mathbf{x}, \bar{\mathbf{x}}\right) = \|\mathbf{x} - \bar{\mathbf{x}}\|. \tag{34}$$

On the other hand, the seminorm $\|\mathbf{x} - \bar{\mathbf{x}}\|$ is equal to the hypothesis margin, see Equation (5):

$$
\begin{aligned}
\|\mathbf{x} - \bar{\mathbf{x}}\| &= \left\|\mathbf{w}^* - \frac{1}{2}\left(\mathbf{w}_* + \mathbf{w}^*\right)\right\|, \\
&= \frac{1}{2}\|\mathbf{w}^* - \mathbf{w}_*\|, \\
&= \frac{1}{2}\left(\|\mathbf{w}^* - \mathbf{w}_*\| - \|\mathbf{w}^* - \mathbf{w}^*\|\right), \\
&= \frac{1}{2}\left(\|\mathbf{x} - \mathbf{w}_*\| - \|\mathbf{x} - \mathbf{w}^*\|\right), \\
&= \text{margin}_h\left(\{\mathbf{x}\}, \mathcal{W}\right). \tag{35}
\end{aligned}
$$

Combining Equation (34) and Equation (35), we conclude $\text{margin}_s\left(\{\mathbf{x}\}, \mathcal{W}\right) \leq \text{margin}_h\left(\{\mathbf{x}\}, \mathcal{W}\right)$ and together with Equation (7) the equality:

$$\text{margin}_s\left(\{\mathbf{x}\}, \mathcal{W}\right) = \text{margin}_h\left(\{\mathbf{x}\}, \mathcal{W}\right).$$

This proves the tightness of the second lower bound and, therefore, the corollary. $\qquad\square$

## D  Experimental setup

This section presents all the required information for reproducing the results of Section 5. Additionally, we give a detailed explanation of how we selected the state-of-the-art methods.

### D.1  Configuration, training, and evaluation of the selected NPCs

**Model selection**   In the experimental evaluation in Section 5, we considered three different NPCs: GLVQ, GTLVQ, and RSLVQ. All methods belong to the family of LVQ algorithms. We focused on theses methods because they optimize the prototypes as fully adjustable parameters and not merely select the prototypes from a given set of training points (e. g., k-nearest neighbors methods). By having access to the full data space and not only the data samples, the methods usually achieve better classification performances with fewer prototypes than prototype *selection* approaches. Below, we give a brief rationale for why we selected the specific LVQ approaches for the presented comparison.

**GLVQ**  From the large LVQ family, GLVQ is by far the most commonly used variant. Therefore, the analysis of GLVQ provides results that are important for many applications. Moreover, the GLVQ loss is the standard loss function to train other LVQ variants.

**GTLVQ**  Compared to NNs, GTLVQ is the closest in terms of CTE and was shown to be robust against adversarial attacks before [26]. Besides, GTLVQ generalizes LVQ to the case of an infinite number of prototypes.

**RSLVQ**  As RSLVQ is not trained using a triplet loss and violates the seminorm assumption, the theorems discussed in Section 4 suggest that adversarial robustness cannot be guaranteed. Including RSLVQ in the comparison, allows for validation of this statement. Additionally, by being trained using the cross-entropy loss, RSLVQ is a variant of LVQ that is similar to NNs.

**Initialization**   The prototypes of the GLVQ and RSLVQ networks were initialized by computing class-wise a k-means, where the number of means was equal to the number of prototypes in the respective class. For GTLVQ, we applied the following standard initialization strategy: The translation vectors are initialized by the same method as used for GLVQ. After that, we initialized each basis $\mathbf{B}_k$ using the following procedure:

1. Determine all the training samples of the correct class for which $\mathbf{t}_k$ is the closest prototype vector in terms of the Euclidean distance. Hence, we consider $\mathbf{t}_k$ as a prototype vector of an ordinary LVQ approach and determine all the training samples that belong to the receptive field of $\mathbf{t}_k$.

Table 3: Configuration of the NPCs used in the evaluation. For GTLVQ, we report the number of prototypes per class and the subspace dimension. Moreover, the ReLU loss for the GTLVQ model refers to the loss of Equation (12).

| Norm | Model | MNIST | | CIFAR-10 | |
|---|---|---|---|---|---|
| | | Loss | Prototypes | Loss | Prototypes |
| $L^\infty$ | GLVQ | GLVQ | 128 ppc | GLVQ | 64 ppc |
| – | RSLVQ | cross-entropy | 128 ppc | cross-entropy | 128 ppc |
| $L^2$ | GLVQ | GLVQ | 256 ppc | GLVQ | 128 ppc |
| | GTLVQ | ReLU ($\epsilon = 1.58$) | 10 ppc, $m = 12$ | GLVQ | 1 ppc, $m = 100$ |

2. Compute the $m$ eigenvectors that belong to the $m$ largest eigenvalues of the estimated covariance matrix over these training samples.

3. Use these $m$ eigenvectors as initialization for $\mathbf{B}_k$ and orthonormalize the resulting matrix if necessary.

**Training**    We trained each method by optimizing the reported loss function. In the case of GLVQ with the $L^2$-norm and GTLVQ, we used the loss function with squared Euclidean distances. This avoids the computation of the square root. As discussed at the end of Section 4, this still optimizes for adversarially robust models.

Each method was trained for 1000 epochs without early stopping. The optimizer was Adam [72], with the default setting of the KERAS framework, a batch size of 128, and an initial learning rate of 0.001. During training, we monitored the validation loss and automatically adjusted the learning rate accordingly. If the validation loss did not decrease over 10 epochs, we reduced the learning rate by a factor of 0.9.

The datasets were normalized to the unit interval. During training, we applied basic data augmentations in the form of random shifts of up to $\pm 2$ pixels and random rotations of up to $\pm 15$ degrees.

**Selected hyperparameters**    In Table 3, the hyperparameter settings for the NPCs are presented. To determine the number of prototypes for GLVQ and RSLVQ, we performed a grid search where we tested the following number of prototypes per class: 1, 2, 4, 8, 16, 32, 64, 128, and 256. The final models have been selected by the smallest URTE for GLVQ and the smallest CTE for RSLVQ. For GTLVQ trained on MNIST, we used the configuration reported by Saralajew et al. [26]. In contrast, the CIFAR-10 model of GTLVQ was selected by the following strategy: We defined the number of prototypes per class to be one and varied the subspace dimension from 1 to 256. For each configuration, we initialized the model and calculated the training accuracy. The final model was selected as the model where the usage of a higher subspace dimension showed no improvement of the accuracy.

Additionally, we considered both the ReLU loss function, see Equation (12), and the GLVQ loss function, see Equation 2, for GLVQ and GTLVQ, while we only considered the cross-entropy loss for RSLVQ. The free parameter of the ReLU loss was always set to be equal to the $\epsilon$ of the $\epsilon$-limited attacks against which the model should be robust.

**Configuration of the PGD attack**    In line with earlier work [33, 39], we used the PGD attack [41] to evaluate the empirical robustness of NPCs. We ran the PGD attack for 200 iterations and used random starts with Gaussian noise. For each sample, the worst-case adversary was selected from three starts. We found no evidence that performing more than three random starts had a significant effect on the reported LRTE.

**Hardware and software frameworks used**    The implementation of the adversarial attack was supplied by the Python FOOLBOX[5] library (version 2.4.0). All models were implemented using the

Python KERAS[6] library (version 2.2.4) with TENSORFLOW[7] back end (version 1.12.0). Evaluation and training were performed on an NVIDIA Tesla V100 32 GB GPU. However, the inference time of each model was obtained using an NVIDIA RTX 2080 Ti GPU. By using a commonly used GPU for this purpose, we hope to make a simple comparison possible.

### D.2 Selection criteria for the state-of-the-art methods

Several important considerations were made when selecting the state-of-the-art certification and verification methods for the comparison. First of all, we wanted to make a comparison with the same number of verification and certification methods. Both approaches play an important role in the research of guaranteed adversarial robustness and therefore require equal consideration. However, as we explicitly focus on fast computation of the guaranteed robustness, a stronger focus was placed on certification methods when considering $L^2$-norm limited attacks. The STN and Smooth methods for certification were selected based on their current state-of-the-art status among certification approaches. As already mentioned, these methods are not deterministic and require extensive sampling. Therefore, the derived certificates differ strongly from the proposed adversarial robustness certificates of NPCs. For this reason, CAP was included as a method more similar to the certification of NPCs. Both IBP and RS were chosen as verification methods because of their focus on fast verification and robustification. RT was chosen to add a second example of guaranteed adversarial robustness outside of NNs in addition to NPCs.

For the comparison of the results, we decided to present for each method the model with the best guaranteed robustness. We chose not to consider ensemble or cascade models (apart from RT) because these models accept a computational overhead for better performances. Considering our scope of fast adversarial robustness certification, we deemed this inappropriate. We presented the CTE, LRTE (if available), and URTE for each method, as reported in the respective papers. This was made possible by the widespread use of the PGD attack as a measure of empirical robustness.

## E   Additional experimental results for Section 5

In the following, we present additional experimental results for Section 5, including an evaluation of the presented NPCs for other $\epsilon$-limited adversarial attacks, an analysis of the NPCs robustness regarding several norm-based adversarial attacks, and an extended discussion of the presented adversarial rejection strategy results.

### E.1   Extended robustness evaluation for Section 5

In this section, we extend the robustness evaluation presented in the comparison of Section 5. For the $L^\infty$-norm, we present results for additional threshold values $\epsilon$, see Table 4. The models were trained and selected like the models presented in the main results. We also include the verification and LRTE results for CAP provided by Gowal et al. [33] for the evaluation of IBP. These results are denoted as CAP-IBP. Furthermore, we included the results of evaluating a 1-nearest neighbor classifier (denoted as 1-NN) by Wang et al. [58], see $L^\infty$-norm with $\epsilon$ equal to 0.1.[8] For the $L^2$-norm, we extend the main comparison with RSLVQ, see Table 5.

In contrast to adversarial examples under the $L^\infty$-norm, no real standardized attack for evaluating the empirical robustness under the $L^2$-norm is available. For the $L^\infty$-norm, the PGD attack plays this role. To provide some insights into the empirical robustness of NPCs trained to optimize $L^2$-norm adversarial robustness, we present in Table 5 the results of an evaluation using the Carlini & Wagner (C&W) attack [14]. There, we present the LRTE obtained with the C&W attack for the GLVQ, RSLVQ, and GTLVQ models and compare it with the reported results of the STN certification method.

We used the C&W attack implementation of the Python FOOLBOX library (version 2.4.0). To determine the trade-off parameter—determining the trade-off between misclassification and perturbation distance in the C&W attack algorithm—a binary search with 10 steps was performed. The attack

Table 4: Comparison of NPCs trained with the $L^\infty$-norm against state-of-the-art methods. Dashes "–" indicate that the quantity is not calculable or reported.

| Dataset | $\epsilon$ | Class | Model | CTE [%] | LRTE [%] | URTE [%] | Notes |
|---|---|---|---|---|---|---|---|
| MNIST | 0.1 | Certify | GLVQ | 3.66 | 6.42 | 6.67 | GLVQ loss, 128 ppc |
| | | | RSLVQ | 1.70 | 93.97 | – | cross-entropy, 128 ppc |
| | | | 1-NN | 3.41 | 27.06 | 27.06 | [58, Table 3] |
| | | | CAP | 1.08 | – | 3.67 | [12, Table 2 "Large"] |
| | | Verify | CAP-IBP | 1.08 | 2.89 | 3.01 | [33, Table 4] |
| | | | RS | 1.32 | 4.87 | 5.66 | [39, Table 3 "RS+"] |
| | | | IBP | 1.06 | 2.11 | 2.23 | [33, Table 4] |
| | 0.2 | Certify | GLVQ | 3.66 | 10.63 | 11.53 | GLVQ loss, 128 ppc |
| | | | RSLVQ | 1.70 | 100.00 | – | cross-entropy, 128 ppc |
| | | Verify | CAP-IBP | 3.22 | 6.93 | 7.27 | [33, Table 4] |
| | | | RS | 1.90 | 6.86 | 10.21 | [39, Table 3 "RS+"] |
| | | | IBP | 1.66 | 3.90 | 4.48 | [33, Table 4] |
| | 0.3 | Certify | GLVQ | 3.66 | 16.39 | 20.58 | GLVQ loss, 128 ppc |
| | | | RSLVQ | 1.70 | 100.00 | – | cross-entropy, 128 ppc |
| | | | RT | 2.68 | 12.46 | 12.46 | [50, Table 3] |
| | | | CAP | 14.87 | – | 43.10 | [12, Table 2 "Small"] |
| | | Verify | CAP-IBP | 13.52 | 26.16 | 26.92 | [33, Table 4] |
| | | | RS | 2.67 | 7.95 | 19.32 | [39, Table 3 "RS+"] |
| | | | IBP | 1.66 | 6.12 | 8.05 | [33, Table 4] |
| CIFAR-10 | 2/255 | Certify | GLVQ | 59.35 | 65.50 | 65.52 | GLVQ loss, 64 ppc |
| | | | RSLVQ | 54.71 | 89.54 | – | cross-entropy, 128 ppc |
| | | | CAP | 31.72 | – | 46.11 | [12, Table 2 "Resnet"] |
| | | Verify | CAP-IBP | 36.01 | 45.11 | 49.96 | [33, Table 4] |
| | | | RS | 38.88 | 51.08 | 54.07 | [39, Table 3 "RS+"] |
| | | | IBP | 29.84 | 45.09 | 49.98 | [33, Table 4] |
| | 8/255 | Certify | GLVQ | 59.35 | 79.54 | 79.62 | GLVQ loss, 64 ppc |
| | | | RSLVQ | 54.71 | 99.04 | – | cross-entropy, 128 ppc |
| | | | RT | 58.46 | 74.69 | 74.69 | [50, Table 3] |
| | | | CAP | 71.33 | – | 78.22 | [12, Table 2 "Resnet"] |
| | | Verify | CAP-IBP | 71.03 | 78.14 | 79.21 | [33, Table 4] |
| | | | RS | 59.55 | 73.22 | 79.73 | [39, Table 3 "RS+"] |
| | | | IBP | 50.51 | 65.23 | 67.96 | [33, Table 4] |

used the Adam optimizer with a learning rate of 0.05 during the 1000 optimization steps, except for RSLVQ, which ran for 3000 steps.

**Results $L^\infty$-norm**   With respect to the additional $\epsilon$ values, the comparison sways in favor of the NPCs for larger $\epsilon$ values as can be seen in Table 4 for the $L^\infty$-norm: For $\epsilon = 0.1$ and $\epsilon = 0.2$ on MNIST, the URTE of GLVQ is higher than the URTE of the certification method CAP and is significantly higher than the robustness guarantees of the verification methods. This is no longer the case for $\epsilon = 0.3$, GLVQ now improves over CAP and is closer to verification methods with regard to URTE. Since large $\epsilon$-limited attacks are a more realistic attack model, we deem this to be an advantageous behavior. For the CIFAR-10 dataset, the same trend is observable—regarding URTE, GLVQ is closer to CAP and the verification methods for $\epsilon = 8/255$ than for $\epsilon = 2/255$.

The same holds for CAP-IBP concerning both the URTE and LRTE. Most notably, with $\epsilon = 0.3$ on MNIST, GLVQ outperforms CAP-IBP by a large margin in terms of LRTE. Despite being verified using an exact method, this model was still trained to optimize certification. Hence, we can conclude that in terms of LRTE, NPCs provide exceptional empirical adversarial robustness compared to other certification methods.

Table 5: Comparison of NPCs trained with the $L^2$-norm against state-of-the-art methods. Dashes "–" indicate that the quantity is not calculable or reported. Values denoted with $^*$ were estimated from figures from the original publication.

| Dataset | $\epsilon$ | Model | CTE [%] | LRTE [%] | URTE [%] | Notes |
|---|---|---|---|---|---|---|
| MNIST | 1.58 | GLVQ | 4.19 | 32.42 | 65.61 | GLVQ loss, 256 ppc |
| | | GTLVQ | 2.92 | 25.86 | 55.32 | ReLU loss, 10 ppc, $m = 12$ |
| | | RSLVQ | 1.70 | 84.26 | – | cross-entropy, 128 ppc |
| | | CAP | 11.88 | – | 55.47 | [12, Table 4 "Large"] |
| | | STN | 1.10 | 10* | 31.00 | [49, Table 1] |
| CIFAR-10 | 36/255 | GLVQ | 51.41 | 58.40 | 61.90 | GLVQ loss, 128 ppc |
| | | GTLVQ | 40.53 | 49.02 | 55.96 | GLVQ loss, 1 ppc, $m = 100$ |
| | | RSLVQ | 54.71 | 75.31 | – | cross-entropy, 128 ppc |
| | | CAP | 38.80 | – | 48.04 | [12, Table 2 "Resnet"] |
| | | STN | 19.50 | 30* | 34.40 | [49, Table 1] |
| | | Smooth | 18* | – | 27* | [18, Figure 5c] |

Table 6: Certified robustness for the $L^\infty$-norm of NPCs trained with the $L^2$-norm.

| Dataset | $\epsilon$ | Model | CTE [%] | LRTE [%] | URTE [%] | Notes |
|---|---|---|---|---|---|---|
| MNIST | 0.1 | GLVQ ($L^2$) | 4.19 | 18.45 | 97.15 | GLVQ loss, 256 ppc |
| | | GTLVQ | 2.92 | 13.98 | 100.00 | ReLU loss, 10 ppc, $m = 12$ |
| | 0.3 | GLVQ ($L^2$) | 4.19 | 89.47 | 100.00 | GLVQ loss, 256 ppc |
| | | GTLVQ | 2.92 | 99.2 | 100.00 | ReLU loss, 10 ppc, $m = 12$ |
| CIFAR-10 | 2/255 | GLVQ ($L^2$) | 51.41 | 64.45 | 74.46 | GLVQ loss, 128 ppc |
| | | GTLVQ | 40.53 | 59.69 | 78.10 | GLVQ loss, 1 ppc, $m = 100$ |
| | 8/255 | GLVQ ($L^2$) | 51.41 | 84.21 | 95.96 | GLVQ loss, 128 ppc |
| | | GTLVQ | 40.53 | 90.2 | 99.61 | GLVQ loss, 1 ppc, $m = 100$ |

In comparison to the 1-NN classifier results, which is also an NPC, GLVQ has a much better adversarial robustness even though the CTE of GLVQ is slightly worse. However, it must be noted that the 1-NN method was not purposefully trained to be adversarial robust.

**Results $L^2$-norm**  Regarding RSLVQ and the $L^2$-norm, we find the same results as presented in the paper for the $L^\infty$-norm: RSLVQ is not robust against adversarial attacks, see Table 5. However, the LRTE score is not trivial. On the other hand, the NPCs that optimize a triplet loss—GLVQ and GTLVQ—are robust against adversarial examples generated by the C&W attack. It must, however, be noted that the LRTE of NPCs is significantly higher than for state-of-the-art methods.

### E.2 Multiple seminorm robustness: Evaluation of the $L^\infty$-norm robustness for $L^2$-norm robustified models

As discussed in Section 6, we can use Hölder's inequality to extend the robustness guarantees between different $L^p$-norms. In this section, we present some preliminary results demonstrating this. Given the NPCs trained using the $L^2$-norm, as presented in Section 5 and Section E.1, Table 6 presents their adversarial robustness with regard to the $L^\infty$-norm, both empirical and guaranteed.

**Results**  The NPCs trained to classify the MNIST dataset have trivial guaranteed robustness and poor empirical robustness. However, if we consider the CIFAR-10 dataset, we find different results. For small values of $\epsilon$, both the empirical and guaranteed robustness against adversarial examples are nontrivial. Interestingly, GLVQ and GTLVQ have a better LRTE with respect to the $L^\infty$-norm when trained using the $L^2$-norm than GLVQ has when trained using the $L^\infty$-norm directly—as can be seen by comparing the LRTE of GLVQ on CIFAR-10 in Table 4 with the LRTE of GLVQ and GTLVQ on CIFAR-10 in Table 6 (the latter has lower LRTE than the former).

Table 7: The first 10 falsely rejected samples from the MNIST test dataset.

| id | $c(\mathbf{x})$ | $c_{\mathcal{W}}^*(\mathbf{x}) = c(\mathbf{w}^*)$ | $c(\mathbf{w}_*)$ | $\mathbf{x}$ | $\mathbf{w}^*$ | $\mathbf{x}^*$ | $\mathbf{w}_*$ | $\mathbf{x}_*$ |
|---|---|---|---|---|---|---|---|---|
| 63 | 3 | 2 | 3 | | | | | |
| 92 | 9 | 9 | 4 | | | | | |
| 115 | 4 | 6 | 9 | | | | | |
| 124 | 7 | 4 | 7 | | | | | |
| 125 | 9 | 9 | 4 | | | | | |
| 149 | 2 | 2 | 9 | | | | | |
| 151 | 9 | 9 | 2 | | | | | |
| 193 | 9 | 9 | 4 | | | | | |
| 195 | 3 | 3 | 9 | | | | | |
| 219 | 5 | 5 | 3 | | | | | |

## E.3 Extended adversarial rejection evaluation for Section 5

In Table 7, we show the first 10 falsely rejected samples from the MNIST test dataset by the adversarial rejection strategy presented in Section 5. For each sample, we visualized the following images: the sample $\mathbf{x}$, the closest prototype $\mathbf{w}^*$, a closest representation $\mathbf{x}^*$ of the closest prototype $\mathbf{w}^*$ in the training dataset, the closest prototype $\mathbf{w}_*$ from another class, and a closest representation $\mathbf{x}_*$ of the closest prototype $\mathbf{w}_*$ from another class in the training dataset. Additionally, we present the following information: the sample identifier (id) in the MNIST test dataset, the true label $c(\mathbf{x})$ of the sample $\mathbf{x}$, the predicted class label $c_{\mathcal{W}}^*(\mathbf{x})$ by GLVQ, and the class label $c(\mathbf{w}_*)$ of the prototype $\mathbf{w}_*$. Note that the predicted class label $c_{\mathcal{W}}^*(\mathbf{x})$ is equal to $c(\mathbf{w}^*)$ and that the samples of class $c(\mathbf{w}_*)$ are the closest samples that can alter the prediction $c_{\mathcal{W}}^*(\mathbf{x})$. Consequently, if we consider $\mathbf{x}$ as a potentially adversarially manipulated sample, the class $c(\mathbf{w}_*)$ is the class that contains the original sample (the non-adversarially manipulated sample) with the highest probability.

We used a rejection threshold of 0.1. Therefore, we rejected all samples with a hypothesis margin of less than 0.1. The rejection threshold reflects the amount of perturbation that can be expected to be used by a malicious attacker without being detectable by other means. According to the original definition of adversarial examples (being perturbed by an imperceptible amount of noise to humans), the threshold can also be considered as the amount of noise that would be *imperceptible* to the human eye. The threshold of 0.1 equates to roughly 6 % of the examples from the MNIST test dataset being falsely rejected.

**Results** We can make several distinct considerations within the set of falsely rejected samples. First of all, several samples are rejected despite being classified correctly by the NPC. Sample 92, for example, is classified correctly as a nine and is still rejected. Second, we find some samples that are

Table 8: Comparison of NPCs trained with the $L^\infty$-norm against robust boosted decision trees and robust boosted decision stumps on tabular data. For Stumps-A, we took the versions trained with the exact robust loss. The URTEs for Stumps-C and RT-C are robust test errors and therefore the best possible URTEs.

| Dataset | $\epsilon$ | Model | CTE [%] | URTE [%] | Notes |
|---------|-----------|-------|---------|----------|-------|
| breast-cancer | 0.3 | Stumps-A | 5.1 | 10.9 | [50, Table 1] |
| | | Stumps-C | 8.8 | 16.8 | |
| | | RT | 0.7 | 6.6 | [50, Table 2] |
| | | RT-C | 0.7 | 13.1 | |
| | | GLVQ | 0.0 | 7.3 | ReLU loss ($\epsilon = 0.45$), 7 ppc, learning rate 0.005, batch size 8 |
| diabetes | 0.05 | Stumps-A | 27.3 | 31.8 | [50, Table 1] |
| | | Stumps-C | 23.4 | 30.5 | |
| | | RT | 27.3 | 35.7 | [50, Table 2] |
| | | RT-C | 22.1 | 40.3 | |
| | | GLVQ | 25.3 | 31.8 | GLVQ loss, 4 ppc, learning rate 0.0002, batch size 64 |
| cod-rna | 0.025 | Stumps-A | 11.2 | 22.6 | [50, Table 1] |
| | | Stumps-C | 11.6 | 23.2 | |
| | | RT | 6.9 | 21.4 | [50, Table 2] |
| | | RT-C | 10.2 | 24.2 | |
| | | GLVQ | 7.8 | 21.4 | ReLU loss ($\epsilon = 0.05$), 8 ppc, learning rate 0.01, batch size 256 |

misclassified and rejected. For obvious reason, these false rejections are not as severe as the rejection of correctly classified samples. Within this category, a further division can be made between samples for which the class $c(\mathbf{w}_*)$ is the ground truth class (e. g., sample 63) and samples for which this is not the case (e. g., sample 115).

## F  Robustness evaluation on tabular data

Additionally to the results on image datasets, we present a robustness evaluation on tabular data and compare it with state-of-the-art robust boosted tree and stump methods. The datasets and the threat models ($\epsilon$ parameters) were chosen in accordance with the evaluation of robust boosted tree-based methods [50]. In particular, we used the source code provided by Andriushchenko and Hein [50] to load, preprocess, and split the following datasets (all available in the LIBSVM library [73]):

**breast-cancer**  a binary classification dataset (also denoted as Breast Cancer Wisconsin (Original) dataset [74]) consisting of 546 training and 137 test samples with 10 features after removing samples with missing values [75];

**diabetes**  a binary classification dataset (also denoted as Pima Indians Diabetes dataset) consisting of 614 training and 154 test samples with 8 features [76];

**cod-rna**  a binary classification task consisting of 59535 training and 271617 test samples with 8 features [77].

Compared to the image datasets that are used in the main part of the paper, these datasets are low-dimensional and do not provide (except for cod-rna) a large number of training samples.

For each dataset, we trained a GLVQ classifier with the $L^\infty$-norm with the setting specified in Table 8. These settings are the result of a hyperparameter optimization by grid search on the specified parameters. With the respective setting, each model was trained several times. From all training runs, the model with the best URTE was selected. In general, the models were trained by the following setting:

- initialization by applying class-wise a k-means algorithm;
- optimizing the models by ADAM with a learning rate scheduler (similar to the one used for the image datasets) and without augmentation;
- stopping the training after 1000 training epochs.

In addition to RT models (see Section 5), we compare the achieved performances with:

- robust boosted decision stumps (denoted by Stumps-A) of Andriushchenko and Hein [50],
- robust boosted decision trees (denoted by RT-C) of Chen et al. [51], and
- robust boosted decision stumps (denoted by Stumps-C) of Chen et al. [51].

For all models, we used the results reported by Andriushchenko and Hein [50]. All models, except for GLVQ, are ensembles.

In Table 8, we report the results of the experiments. Summarily, there is no superior method. For example, on the breast-cancer dataset, the RT achieves the best URTE, but GLVQ has a better accuracy and is just slightly worse in terms of URTE. On the other hand, the Stumps-C outperforms all other models on the diabetes dataset with respect to URTE, but RT and GLVQ are close behind it. Moreover, on the cod-rna dataset, the RT and the GLVQ classifier achieve the same and best URTE and outperform all other models. With no clear superior method for either guaranteed robustness or clean accuracy, GLVQ can therefore be considered as state of the art for guaranteed adversarial robustness on tabular datasets, together with the stump- and tree-based methods.