[Reviews · NeurIPS 2020]

Review 1

Summary and Contributions: This work studies the robustness of Nearest Prototype Classifiers models. The authors prove that the hypothesis margin of the model serves as a tight lower bound for the robustness certification. The author also proposed a method to improve the robustness of the NPC models.

Strengths: The adversarial robustness is a hot research topic in deep learning. However, the robustness problems for other machine learning models are not so well-studied. Recent, there is an increasing number of researchers started to study other models' robustness. In this work, the authors provide an intuitive lower bound for the robustness of NPC models, in Theorem 2, which is the hypothesis margin. Due to the simplicity of the hypothesis margin given by Theorem 1, the authors are able to formulate a robust training loss for NPCs which encourages large hypothesis margins.

Weaknesses: In Table 1, the LRTE of RSLVQ is so large (100% or 99%) that it seems to be not robust at all. I am not sure if robust training of RSLVQ really works on MNIST or CIFAR. GLVQ cannot beat existing methods, either. In general, I suspect that NPC methos are not popular for image datasets. So the poor performance is expected. Maybe the authors can test on some smaller tabular data with lower features dimension, where I think the results are going to be better. I don't think it is necessary to compare NPCs with DNNs because they are designed for different tasks.

Correctness: I checked the proof of theorem 1 and 2 with some easy cases and it seems to be correct.

Clarity: The paper is well written. However, I think in general the paper is hard to follow, due to nuerous mathematical definitions and terms. I understand the authors want to make it mathematically rigorous. But I suggest the authors to add some figures and examples for theorem 1 and 2. Especially, I think both Theorem 1 and Theorem 2 can be easily illustrated by an example on R^2 with L2 norm. This will help the readers to quickly get the intuitions of the ideas.

Relation to Prior Work: The authors clearly discussed how this work differs from previous works. However, there are some missing citations that I think the authors may need to discuss. All these works are on the robustness of non-deep learning models, such as kNN and trees. Yang et al, "Robustness for Non-Parametric Classification: A Generic Attack and Defense", AISTATS 2020. Wang et al, "Evaluating the Robustness of Nearest Neighbor Classifiers: A Primal-Dual Perspective" arXiv 2019. Chen et al, "Robustness Verification of Tree-based Models", NeurIPS 2019. Wang et al, "On lp-norm Robustness of Ensemble Decision Stumps and Trees", ICML 2020.

Reproducibility: Yes

Additional Feedback: I think the authors may also test some easy benchmarks to show that the proposed method really works. For example, if the original natural training is used, but instead of using 1-nearest neighboring rule, we use k-nearest neighboring rule in inference, does the model become more robust? Or we can adopt the idea of randomized smoothing in Cohen et al ICML'19, if at inference time, we random sample some points in the neighborhood of the test example and use the majority of their labels as our final label. This may also improve model robustness.


Review 2

Summary and Contributions: This paper studies the robustness certification problem of nearest prototype classifiers (NPCs). NPCs are related to 1-nearest neighbour (1-NN) classifiers, and it uses a similar classification rule as 1-NN. However, in NPCs we don't need to keep the full dataset, but only a few prototype vectors. This paper studies the lower bound of classification margin (which determines model robustness) under this setting.

Strengths: The theorems in the paper are technically sound. I did not check the proofs in detail but the results look reasonable. This is also the first paper that specifically studies the robustness certification problem of NPCs. The authors conduct empirical studies on a few variants of NPCs (GLVQ, RSLVQ, GTLVQ) and found that some of the variants are more robust than others. This can be a useful observation when a robust NPC is desired.

Weaknesses: The robustness certification of K-nearest neighbor classifiers have been studied in previous works ([1], which is not cited). Once the prototypes are fixed after training and distances between all pairs of points are obtained, it seems NPCs have little difference comparing to nearest neighbor classifiers in the robustness certification process. [1] studies robustness certification of K-NN classifiers with any Lp norm as distance metric. This paper seems to be very related to robustness certification of 1-NN. The authors should discuss the connections to [1]. Can the results in [1] be extended to NPCs? The paper is not well motivated. Does NPC provide any unique benefits comparing to other approaches? Is NPC necessary for any critical applications? In Table 1 it seems NPC based methods are not competitive comparing to other approaches (e.g., IBP, which is also very efficient). The paper claims that (e.g. in line 319) the robustness guarantee of NPCs surpassed other baselines, but the experiments do not seem to support this claim.

Correctness: Some minor issues: In Table 1, the table does not include the computational cost (in the number of forward passes). This should be included if possible. In Table 1, I am unclear the "class" (certify/verify) is meaningful. In fact, all four non-NPC methods (CAP, RS, IBP, RT) are both robustification and verification methods. These methods all include a training process to produce a tight certification bound.

Clarity: Overall this paper is well presented.

Relation to Prior Work: The authors need to discuss previous works on robustness certification of nearest neighbor classifiers.

Reproducibility: Yes

Additional Feedback: Overall this paper is well presented and technically sound. However, I believe its technical contribution is minor and it does not have significant impact to this field. Thus I vote for a weak reject. To increase the contribution of this paper, the authors can consider designing training algorithms that improves the provable robustness of NPCs. For example, RSLVQ is a strong method (in Table 1 it achieves very competitive clean test error); can we improve its robustness to the same level of other baselines? References: [1] Wang, Lu, et al. "Evaluating the Robustness of Nearest Neighbor Classifiers: A Primal-Dual Perspective." ------------ After rebuttal reviews: Thank you for pointing out the applications of NPCs to medical research - I was unaware of that. I looked for the paper in The Lancet but I did not find anywhere they mention GMLVQ nor NPC. I might missed something because it is a paper in a completely different field, but when the authors are preparing the new version it is better to mention explicitly where and how NPC are used. From the analysis point of view, although this paper claims to work for any semi-norm, it does not seem to me that the differences between Lp norm and semi-norm are that large during the analysis - the paper on kNN classifier for Lp norm should be easily extend to 1-NN and seminorm as well, I believe. For the next revision of this paper, it is necessary to discuss how the analysis is different from the previous work, and why seminorm handling is challenging. It will enhance the technical contribution of this paper. Reviewer #1 pointed a few papers on robust decision tree based models, which are quite useful. I like the newly provided experiments on tabular datasets in the author response. For NPCs I think it makes more sense to work on tabular data, and in the settings where we need more explanability. I don't think NPC, nearest neighbour or tree based models should be compared directly against deep neural networks on vision datasets. I think the authors can include more comprehensive discussions in related work on nearest neighbour or tree based methods, and include more datasets like in the tree papers (Chen et al., Wang et al.,) pointed by reviewer #1. Using datasets like CIFAR or comparing directly to neural networks is probably not a good fit here, and they will only make the results look worse. Overall, I still put the paper in the borderline, I think it is looks correct and the results are new, but compared to existing works on kNN classifiers the contribution is simple and the applications of NPCs are limited. Thus the paper only has limited impact. Also, the experiments are mostly on vision datasets which are not the main applications of NPC. This paper is okay to accept, but I won't feel disappointed if it is rejected by the AC.


Review 3

Summary and Contributions: This paper consists of two parts: (1) it first analyzes adversarial robustness on the "nearest prototype classifier" (NPC) and (2) it then robustifies the NPC with the derived hypothesis margin. The experiments show that NPC do not outperform state-of-the-art robustly trained DNNs but NPC could have potential advantage for on-line non-adversarial examples detections.

Strengths: - the math part of the paper is relatively easy to follow - the novelty of the work is to identify a new ML model that could have easy-to-derive lower bound of robustness certificate (i.e. the hypothesis margin in the paper) and have fine performance on mnist and cifar dataset. - the margin computation (Thm 1) has relatively low computation cost (if the size of the prototypes W is not large) and thus it's possible to do on-line non-adversarial detections

Weaknesses: - there are several places that have inaccurate descriptions or misleading: e.g. For IBP [29], the method is actually "certification method" because it introduces the interval bounds in the training. It is not based on the "verification" method. They use Interval bounds in the training, and some of their results use MIO verifier to evaluate the best test errors they can get. Also, the computation complexity of [12, 17, 18] in Line 27 are totally different. Some are polynomial time, some are NP-complete. Usually in this field, only the formal verification based method such as [17] will be described as computationally expensive. For line 29, those methods are not used for detecting adversarial examples, because they provide a certified region for consistent classifications as the input example x. They can only be used to detect guaranteed "non"-adversarial example of x given a new input x'. - there are also some places that are not clear: e.g. it's not clear how to train the prototypes w in equation 2. It looks like d is pre-defined and the only parameters are w. Also, what are the relations of eq 3 to eq 2? - The NPC models are not as commonly used as other models such as NN in my understanding, so it's not clear how useful/important the robustness analysis is in this regard.

Correctness: In line 196-200, what is the definition of original sample? If margin_h > eps, then we can say that there'll be no adversarial examples within distance eps to a given example x (due to the in-equality 8). But it's not clear why here there are x and x^bar.

Clarity: It's fine.

Relation to Prior Work: See weakness section.

Reproducibility: No

Additional Feedback: 1. What't the motivation of using NPC given it's not the mainstream models? 2. Why in Table 1, there are some places have "-"? What does ppc mean? Is that the number of prototype? 3. Please elaborate more the training of NPC 4. Line 222-224, it's not clear about the reasoning. Please elaborate more. 5. Line 275: "the guarantees improves too". Note that the guarantees are "only" for the "training set". There's no guarantees on the test set by looking at the training loss. Hence, the sentence should be rewritten to avoid misunderstanding. 6. For the idea of perturbing parameters in NN as mentioned in Sec 7, there are some results in the below work: Weng etal, Towards Certificated Model Robustness Against Weight Perturbations, AAAI 2020. Can the authors explain what would be the difference of their proposal between this work? ================= post-rebuttal: I've read the authors' rebuttal.


Review 4

Summary and Contributions: The article shows (in a general settings) that classifiers, which are based on distances resp. seminorms, can link the margin to the adversarial robustness, and demonstrates according robustness in a couple of samples. The paper puts a very interesting spotlight on architectures which, recently, became popular for few shot learning, and provides a very simple, though attractive way to check robustness, which comes from basic geometric considerations.

Strengths: Very relevant topic, simple but effective criterion, supported by experiments

Weaknesses: It is unclear why we need semi-norms, can you provide a use case demonstrating the benefit (such as getting better bounds with problem-adapted norm). The margin usually gets bad for high dimensions (so SVM being independent of dimension is somewhat of a cheat), can you comment on that? Also, there seems to exist prior work (also by the authors), which makes the result a bit incremental since the relevant base cases seem to be subject to prior work already. The experiments do not yet include complex real life data (where, typically, the dimensionality is much bigger, albeit not the intrinsic one. But the data manifold is often very thin, such that I wonder whether the results are good enough to transfer to complex settings.)

Correctness: Seems correct to me

Clarity: Easy to read and well structured

Relation to Prior Work: There is previous work on robustness of LVQ to adversarial previous to the author's one, albeit in the standard norm, as well as quite some work on robustness of kNN and other methods: Analyzing the Robustness of Nearest Neighbors to Adversarial Examples Yizhen Wang, Somesh Jha, Kamalika Chaudhuri; Interpretable machine learning with reject option; Johannes Brinkrolf, Barbara Hammer; Further, the connection of the hypothesis margin to a difference of distances goes back to Crammer et al in the original form, and could be more clearly appreciated as such.

Reproducibility: Yes

Additional Feedback:

[Author Response · NeurIPS 2020]

*We thank the reviewers for the careful comments, questions, and recommendations that surely will improve the quality of our paper. Below we highlight some common themes within the reviews and finish with some general clarifications.*

**1) Motivation.** After reevaluating our work considering the reviews, we agree that our work neglects to adequately motivate the importance of the study of guaranteed adversarial robustness for NPCs and the role these models play in Machine Learning (ML). Unfortunately, the newly introduced broader impact section enticed us to shift most of the motivation from the introduction to this new section. We will revert this shift in the camera-ready version and also add some additional background information on NPCs here. As discussed in the broader impact section, NPCs are considered to be one of the most interpretable ML models. This makes NPCs a preferred choice in medicine, where models are required to be interpretable for use in clinical trials. This is highlighted in the recent study *Urine steroid metabolomics for the differential diagnosis of adrenal incidentalomas in the EURINE-ACT study: a prospective test validation study,* published in the renowned journal THE LANCET. In this work, GMLVQ is used to construct an extraction analysis method for detecting adrenal tumors. The resulting model is evaluated in one of the largest studies of ML in medicine to date, consisting of over 2000 participants and spanning almost 10 years. In addition to this, as R#4 rightfully pointed out, NPC-like models are often used in the few-shot and meta-learning fields, resulting in several cross-breeds between deep learning models and NPCs (e.g., *Prototypical Networks for Few-shot Learning*).

**2) Related work.** As R#1 and R#2 mentioned, there is a strong relation between NPCs and 1-NN. This relation is discussed in the last related work paragraph, but it will be extended to include work mentioned by the reviewers. *Evaluating the Robustness of Nearest Neighbor Classifiers: A Primal-Dual Perspective* contains a formulation similar to the hypothesis margin, but is limited to $L^1$-, $L^2$-, and $L^\infty$-norms for the attack and $L^2$-norms for the classifier metric. Additionally, their methods are order of magnitudes slower and do not achieve the same robustness as GLVQ. *Towards Certificated Model Robustness Against Weight Perturbations* discusses weight perturbations without the relation to adversarial changes to the input and does not use the margin concept. However, the approaches could be complementary. *Robustness for Non-Parametric Classification: A Generic Attack and Defense* is related to the adversarial robustness of NPCs because of the relation to 1-NN. However, their method is more time consuming and is not studied with respect to seminorm-based classification rules like LMNN. *On lp-norm Robustness of Ensemble Decision Stumps and Trees* and *Robustness Verification of Tree-based Models* are partly covered by the cited papers [46, 47], but they will be included and discussed in the camera-ready version. *Interpretable machine learning with reject option* studies adversarial rejection in a similar manner as our work, but focuses on GMLVQ models trained with reject option.

**3) Additional experiment.** As highlighted by R#1, NPCs are often applied to tabular data. During the experimental design, we performed an additional experiment using tabular datasets more closely related to the usual NPC domain:

| Dataset | $\epsilon$ of $L^\infty$-attack | Robust Stumps | | Robust Trees | | GLVQ with $L^\infty$-norm | |
|---|---|---|---|---|---|---|---|
| | | CTE | URTE | CTE | URTE | CTE | URTE |
| breast-cancer | 0.3 | 5.1% | 10.9% | 0.7% | **6.6%** | **0.0%** | 8.7% |
| diabetes | 0.05 | 27.3% | 31.8% | 27.3% | 35.7% | **22.0%** | **33.7%** |
| cod-rna | 0.025 | 11.2% | 22.6% | **6.9%** | **21.4%** | 7.6% | **21.4%** |

In the experiment, GLVQ is compared to *Provably Robust Boosted Decision Stumps and Trees against Adversarial Attacks* [46]*,* using a similar setup. The comparison shows that an NPC trained using a triplet loss has a guaranteed robustness similar to the robustified stumps and trees. We agree with the reviewer that by selecting the presented experiments to make the comparison to state-of-the-art methods, we, unfortunately, ignore the power of NPCs for tabular data. We will include the additional experiment in the supplementary to counterbalance this.

**4) Clarifications.** First, RSLVQ serves the role of highlighting that an NPC trained without a triplet loss of seminorms does not necessarily become robust. Hence, the atrocious robustness of RSLVQ, as observed by R#1, is therefore the expected/desired behavior—how RSLVQ can be robustified is not clear yet, as questioned by R#2. We will move the overview found in Sec. C.1 from the supplementary to the main paper to clarify the model selection intentions. Similarly, the importance of the seminorm independence will also be amplified. In short, this allows for using task-specific or adaptive metrics. NPCs such as LMNN and GMLVQ are therefore still provably robust against adversarial examples under the seminorm they use for classification—according to the provided theorems. Second, as R#2 correctly observed, all compared methods—except for RSLVQ—are also robustification approaches. With the classification of the used models as either verification or certification, we refer to the method used for obtaining the URTE. Hence, the combination of verification/certification and robustification is possible. Similar, as R#3 mentioned, the robustification of IBP is indeed based on a certification method, as it does not optimize the exact robust loss during training. The URTE of IBP is however obtained using a MIP solver, and therefore the provided URTE falls within the verification class. This nuance will be made more concrete in the final version. Lastly, some short notes: We will add a dedicated section to the supplementary to discuss LVQ and specifically its training process. We will refer to this section in Sec. 3 to answer questions regarding the training of LVQ models. Computational cost/time complexity of the models will be included in the tables. A figure will be added to Sec. 4 to explain the geometrical intuition behind the formulas and proofs. We will clarify and rephrase the sentences mentioned by R#3 and clarify that "–" in Table 1 means not reported or calculable.

[Meta-Review · NeurIPS 2020]

Four experts in the field reviewed the paper. Two reviewers put the paper marginally above the acceptance threshold, and the other two placed it marginally below. The reviewers found the rebuttal useful, but they had lingering questions, such as the work's relationship to 1-NN. Considering the nature of the reviewers' comments, AC felt the concerns could be addressed in a paper revision. The decision is to recommend the paper for acceptance. AC suggested the authors addressing the lingering questions to their best ability when preparing the camera-ready.